# Balance control threshold to vestibular stimuli

Martin Simoneau[1,2] 🆔, Mujda Nooristani[2,3] and Jean-Sébastien Blouin[4] 🆔

[1]*Department of Kinesiology, Faculty of Medicine, Laval University, Quebec City, Quebec, Canada*
[2]*Center for Interdisciplinary Research in Rehabilitation and Social Integration (Cirris), Quebec City, Quebec, Canada*
[3]*School of Rehabilitation Sciences, Faculty of Health Sciences, University of Ottawa, Ottawa, Ontario, Canada*
[4]*School of Kinesiology, The University of British ColumbiaBritish Columbia, Vancouver, Canada*

Handling Editors: Richard Carson & Ross Pollock

The peer review history is available in the Supporting Information section of this article (https://doi.org/10.1113/JP288016#support-information-section).

**Abstract figure legend** (*A*) We exposed participants standing on force plates to sinusoidal electrical vestibular stimulation (EVS) at varying amplitudes (0.2, 0.4, 0.6 mA) and frequencies (0.1, 0.2, 0.5, 1 Hz). (*B*) Given the role of internal head motion estimates in generating balance-correcting responses, we used signal detection theory to quantify non-perceptual balance control thresholds to isolated vestibular stimuli. We compared the mean and variability of the lateral force distribution at determined amplitudes of vestibular stimuli (i.e., multiples of +/− 0.05 mA; B, upper panel). The overlap between the two lateral force distributions was quantified by $d'$, which varied linearly with the EVS signal (B, lower panel). The vestibular-evoked balance control thresholds were computed as the EVS amplitude at which $d' = 1$. (*C*) The vestibular-evoked balance control thresholds ranged from 0.09 to 0.57 mA, and increased with EVS amplitude and decreased with frequency.

**Abstract** Bipedalism renders our erect posture unstable, requiring the integration and processing of multisensory information to remain upright. To understand how each sense contributes to balance, perceptual thresholds to isolated sensory disturbances while standing are typically quantified. Perception, however, is distinct from balance control. Both processes rely on distinct internal body representations, and participants can misattribute the consequences of self-generated balance-correcting actions as an external perturbation. Here, we used signal detection theory to quantify non-perceptual balance control thresholds to isolated vestibular stimuli given the role of vestibular cues in generating balance-correcting responses. We exposed participants standing on force plates to electrical vestibular stimulation (EVS) at varying amplitudes (0.2, 0.4, 0.6 mA) and frequencies (0.1, 0.2, 0.5, 1 Hz). Stimuli delivered at 0.2 mA (0.1–0.5 Hz) and 0.4 mA (0.1, 0.2 Hz) remained unperceived but evoked whole-body responses above the sensorimotor noise underlying balance control. Balance control thresholds ranged from 0.09 to 0.57 mA; they increased with EVS amplitude and decreased with frequency. The physiological mechanisms underlying these EVS amplitude and frequency effects involved a decrease in response gain with increased stimulus amplitude and a reduction in response variability with increased stimulus frequency. Our findings demonstrate that balance responses to isolated vestibular stimuli can be quantified below perceptual thresholds and highlight the dynamic regulation of response gain and the influence of whole-body motion variability in the vestibular control of balance. Our results also open the door to assessing the isolated vestibular contributions to postural control in people with balance impairments.

(Received 30 October 2024; accepted after revision 12 March 2025; first published online 4 April 2025)

**Corresponding author** Martin Simoneau: Centre Interdisciplinaire de Recherche en Réadaptation et en Intégration Sociale (Cirris) du CIUSSS de la Capitale Nationale, Université Laval, 525 boul. Hamel, Québec, QC G1M 2S8, Canada. Email: martin.simoneau@kin.ulaval.ca

**Key points**

- Upright balance control relies on sensory information from multiple sensory systems, but balance control thresholds to isolated sensory stimuli remain largely unknown because these stimuli, or their associated responses, can be perceived.
- We applied isolated electrical vestibular perturbations and used signal detection theory to quantify balance control thresholds to unperceived sensory stimuli.
- Vestibular stimuli delivered at 0.2 mA (0.1–0.5 Hz) and 0.4 mA (0.1 and 0.2 Hz) remained unperceived but evoked balance-correcting responses above the sensorimotor noise underlying the control of standing.
- Balance thresholds increased with current amplitude (0.2–0.6 mA) and decreased with stimulus frequency (0.1–1 Hz) and were linked to decreased gain of lateral force and reduced lateral force variability as current amplitude and frequency increased, respectively.
- These results pave the way for uncovering the sensory contributions to the non-perceptual mechanisms regulating balance-correcting motor commands essential for bipedalism and their potential role in balance impairments.

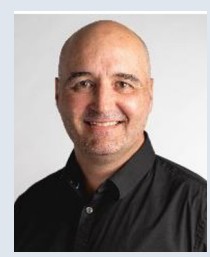

**Martin Simoneau** obtained his PhD in neuromechanics from Laval University in 2000, and thereafter, he was a postdoctoral fellow at Northwestern University for 2 years. After he completed his postdoctoral training, he joined the Faculty of Medicine at Laval University. His research group adopts a multidisciplinary approach, combining experimental methods with signal analysis and modelling. His current research focuses on sensorimotor processes that control human movement and balance and the disorders resulting from damage to the central nervous system. He is particularly interested in how neural processes combine sensory information from the vestibular organs, eyes, muscles and skin to select motor commands and control balance.

## Introduction

The human bipedal posture is mechanically unstable, leading to whole-body oscillations that must be sensed and corrected by the neural networks subserving balance control (Peterka & Benolken, 1995). Sensory end organs in the visual, auditory, somatosensory and vestibular systems detect these small whole-body motions, providing critical information for balance control (Dozza et al., 2007; Peterka, 2002). Despite their essential role, the individual contribution of each sensory system to postural control remains a fundamental enigma to reveal the physiological mechanisms regulating balance-correcting motor commands necessary for bipedalism. One accepted approach to address this question relies on quantifying perceptual thresholds to balance perturbations targeting a specific sensory system (Fitzpatrick & McCloskey, 1994) to characterize the performance limits of a sensory system given its specific physiological noise characteristics. This approach, however, has several limitations in the context of standing balance. First conscious perception of motion happens only when a perceptual threshold is reached (Teasdale et al., 1999), yet we know that most self-generated balance-correcting motor actions occur without conscious perception (Fitzpatrick et al., 1992; Luu, 2010; Luu et al., 2012). Second our balance reactions to small perturbations can be misperceived as an external perturbation (Teasdale et al., 1999; Tisserand et al., 2022; Wardman, Taylor et al., 2003). Finally targeted sensory perturbations evoke whole-body balance responses that may be perceived via motion detected by multiple sensory systems (Teasdale et al., 1999; Tisserand et al., 2022), creating ambiguity regarding the sensory source(s) underlying the perceptual detection process. Here, we addressed these issues by quantifying the threshold of the balance system (as opposed to perceptual thresholds) to perturbations of vestibular origin.

The vestibular end organs encode head linear and angular accelerations in space. Vestibular perceptual thresholds to linear and angular motion are typically 1cm/s and 1°/s, respectively (Benson et al., 1986; Soyka et al., 2012; Valko et al., 2012), with larger stimuli required to perceive stimulus frequencies <1 Hz (Grabherr et al., 2008; Soyka et al., 2011; Soyka et al., 2012). Although vestibular perceptual thresholds (particularly lateral translation) may be correlated to whole-body oscillations while balancing (Karmali et al., 2021), these perceptual studies relied on whole-body perturbations (mostly in seated participants) that cannot be applied to standing participants for targeted vestibular testing. Indeed whole-body motion imposed while upright would lead to self-motion sensed by multiple cross-modal sensors. An alternative method to probe specifically the vestibular control of balance involves applying transmastoidal currents through surface electrodes (Dakin et al., 2007; Fitzpatrick & Day, 2004; Pialasse et al., 2015). This method, termed 'electrical vestibular stimulation' (EVS), modulates the firing rate of primary vestibular afferents (Goldberg et al., 1982; Kwan et al., 2019) without imposing a mechanical head motion. In a binaural bipolar configuration, EVS results in a net vestibular signal of virtual head motion around a vector directed posteriorly, pointing ∼17°–19° above Reid's plane (Day & Fitzpatrick, 2005; Fitzpatrick & Day, 2004; Schneider et al., 2002) with a gravity-dependent inference of linear acceleration (Khosravi-Hashemi et al., 2019). When facing forward the EVS-evoked virtual head motion signals result in balance-correcting responses in the frontal plane that scale with the current amplitude (Latt et al., 2003; Wardman, Day et al., 2003). Consequently when applied above a certain current amplitude, EVS-evoked balance-correcting responses can be perceived (as opposed to the virtual head motion signals; Wardman, Taylor et al., 2003). To avoid issues related to the perceptual detection of imposed perturbations, we used signal detection theory (Green & Swets, 1966) to quantify balance responses that remained unperceived but were evoked by vestibular cues of head motion above the sensorimotor noise underlying the control of natural whole-body oscillations while standing upright.

We quantified the balance thresholds to vestibular stimuli by exposing standing participants to EVS of different amplitudes (i.e. 0.2, 0.4 and 0.6 mA) and frequencies (i.e. 0.1, 0.2, 0.5 and 1 Hz) while measuring the net lateral ground reaction force as a proxy for the net horizontal acceleration of the centre of mass (CoM; Shimba, 1984). A primary aim of the present study was to characterize balance responses to EVS stimuli below perceptual thresholds. Because the larger whole-body movements induced by higher EVS currents (Day et al., 1997; Latt et al., 2003; Wardman, Day et al., 2003) were more likely to be detected by multiple sensors and to potentially evoke (mis)perception of motion (Teasdale et al., 1999; Tisserand et al., 2022), we hypothesized that higher EVS amplitudes would lead to higher balance thresholds to vestibular stimuli. To support this physiological mechanism, we predicted that an increase in lateral force variability to larger EVS stimuli amplitude would explain our results. We further characterized balance thresholds across EVS frequencies (i.e. 0.1, 0.2, 0.5 and 1 Hz). As vestibular afferents (Kwan et al., 2019) and tilt perceptual thresholds (Allred & Clark, 2023; Lim et al., 2017; Wagner et al., 2022) decrease with stimulus frequencies increasing from 0.1 to 1 Hz, we expected smaller balance thresholds to vestibular stimuli as EVS frequency increased. The results from the current experiment supported our hypotheses: balance thresholds to EVS increased with stimulus amplitude and decreased with stimulus frequency. However the

mechanism underlying the increase in balance thresholds to larger vestibular stimuli did not support our prediction: larger balance thresholds were associated with a decrease in the gain of the lateral force as stimulus amplitude increased. We further revealed that the reduction in balance thresholds as a function of EVS frequencies was due to a decrease in the variability in lateral force oscillations at higher frequencies.

## Methods

### Participants

Ten healthy adults [six males and four females, mean age: 27.5 ± 6.7 years, mean (±SD) height: 174.4 ± 7.9 cm, mean mass: 71.1 ± 12.9 kg] with no musculoskeletal or neurological disorder history participated in the experiment. The study protocol complied with the Declaration of Helsinki, except for registration in a public database (clause 35), and was reviewed by the University of British Columbia Clinical Research Ethics Board (H17-02672). All participants gave written informed consent before participating.

### Procedures

During the experiment, participants stood upright barefoot on two force platforms (model OR6-7-1000, AMTI, Watertown, MA, USA) with eyes closed, arms by their sides and heads facing forward. We normalized stance width to each participant's foot length; stance width was defined as the distance between the base of the fifth left and right metatarsals. Using a custom LabVIEW software (National Instruments, Austin, TX, USA), we digitized and recorded the force plate data at 2048 Hz using a multifunction data acquisition board (model PXI-6289, National Instruments). The net lateral force along the frontal plane was calculated by summing the lateral ground reaction forces from both force platforms. We elected to describe whole-body motions using lateral force, as this measure is proportional to the whole-body CoM net horizontal acceleration (Shimba, 1984; Zatsiorsky & King, 1998). We reported the forces applied to the body, with positive (negative) lateral forces representing right (left) whole-body accelerations. Before starting the experiment and at every ~10 trials, we asked participants to close their eyes to feel their natural whole-body motions for ~20 s. This allowed participants to sense their whole-body motion without any additional stimulus. When balancing upright, Wardman, Taylor et al. (2003) reported that participants always perceived the direction of the EVS-evoked sway as opposed to the direction of the imposed perturbations (i.e. they never perceived the direction of the EVS perturbation).

Consequently we determined if participants felt the balance motion induced by EVS by asking them to report if they felt any abnormal whole-body motions after each EVS and sham trial. Perception data were collected from eight participants.

We generated 30-s sinusoidal electrical vestibular stimuli using custom scripts developed with LabVIEW (National Instruments, Austin, TX, USA). Each trial lasted 40 s but included 5-s periods without EVS at the beginning and end of each trial to facilitate data alignment (see later the procedure for aligning both signals). These signals were sent to a constant-current stimulator (model DS5 stimulator, Digitimer Ltd, Welwyn Garden City, UK) through the same data acquisition board that recorded force plate data (model PXI-6289, National Instruments, Austin, TX, USA). To reduce any non-vestibular sensation evoked by electrical stimulation (e.g. cutaneous), we applied an anaesthetic cream (tetracaine HCl gel 4%, Smith & Nephew Medical Ltd, UK) on the mastoid processes before the stimulating electrodes were placed. During the experiment, we asked participants if they felt distracting cutaneous sensations under the electrodes every ~10 trials. Thirty minutes after the anaesthetic cream was applied, we secured two carbon rubber electrodes (9 cm$^2$) coated with gel (Spectra 360, Parker Laboratories, Fairfield, NJ, USA) on the mastoid processes with an elastic band and surgical tape to create a binaural bipolar configuration. Binaural bipolar EVS modulates vestibular afferents from both otolithic and semi-circular canal end organs (Goldberg et al., 1984; Kim & Curthoys, 2004; Kwan et al., 2019), leading to head motion vectors in which their net summation evokes a craniocentric sensation of head roll with an inferred interaural linear acceleration (Day & Fitzpatrick, 2005; Fitzpatrick & Day, 2004; Khosravi-Hashemi et al., 2019). When the head faces forward, EVS produces a virtual head motion signal in the frontal plane detected by the balance system, which evokes whole-body balance-correcting responses. To maximize the vestibular-evoked balance responses along the frontal plane, participants kept their heads forward with an ~17°–19° upward pitch (Fitzpatrick & Day, 2004; Lund & Broberg, 1983; Mian & Day, 2009; Reynolds, 2011; Schneider et al., 2002). An experimenter verified that participants maintained this head orientation by monitoring the position of a laser (mounted to the head using a welder head gear) pointing to a target (~1 m behind the participants). Based on our convention, a positive EVS signal represents an anode right/cathode left current and induced whole-body motion to the right (vice versa for a negative EVS signal).

To verify our hypothesis that higher EVS currents would increase vestibular-evoked balance control thresholds, as larger current amplitudes induce larger whole-body motions that may engage perceptual mechanisms, we determined balance thresholds to sine

wave vestibular stimuli of 0.2, 0.4 and 0.6 mA amplitudes. Additionally because vestibular afferent and tilt perceptual thresholds decrease with higher frequencies (from 0.1 to 1 Hz), we tested whether balance thresholds improved with EVS frequencies of 0.1, 0.2, 0.5 and 1 Hz. Participants were exposed to 14, 7, 3 and 2 trials for the 0.1, 0.2, 0.5 and 1 Hz EVS frequencies, respectively, resulting in 42, 42, 45 and 60 sine wave cycles across EVS frequencies. We also included a control condition (sham EVS trials) where we recorded the lateral forces without an EVS signal (14 trials per condition). We performed this condition to verify that the computed vestibular-evoked balance control thresholds emerged from the association between the EVS and the lateral force signals (as opposed to spurious correlations between these signals due to their statistical properties). We used methods based on signal detection theory previously used for vestibular neuronal thresholds (Jamali et al., 2014; Kwan et al., 2019; Sadeghi et al., 2007) to compute vestibular-evoked balance control thresholds. The detection threshold represents the minimal change in the vestibular stimuli that results in an obvious shift in lateral force distribution (i.e. whole-body horizontal CoM acceleration). Signal detection theory assumes that the lateral force is normally distributed in presence and absence of EVS. We calculated the separation between the lateral force distribution at a given EVS amplitude and when the EVS amplitude was 0 for all EVS conditions. Similar to previous studies quantifying vestibular afferent thresholds (e.g. Sadeghi et al., 2007), we first extracted the lateral force signal at the frequency of the EVS by filtering the force signal using a Gaussian narrow band filter (peak of Gaussian at the EVS frequency with a full width at half maximum of 0.1 Hz). Then, we calculated the phase difference between the EVS and force signals using cross-correlation. We removed this phase difference between both signals (Jamali et al., 2014; Kwan et al., 2019; Sadeghi et al., 2007) and plotted the lateral force as a function of the EVS (Fig. 1*B*, upper panel). We also removed the first cycle of each trial to avoid potential transient effects associated with the onset of EVS. Thus the remaining cycles across EVS frequencies for each stimulus amplitude were 28 for 0.1 Hz, 35 for 0.2 Hz, 42 for 0.5 Hz and 58 for 1 Hz. Due to the larger number of cycles for the 1 Hz EVS condition (58 cycles), we also estimated vestibular-evoked balance control thresholds with 42 cycles. These analyses revealed lower median thresholds with fewer cycles (up to 0.01 mA for 42 cycles). Because of the minimal differences and the expected decrease in vestibular thresholds with higher EVS frequency, we presented data only from the 58 cycles for the 1 Hz EVS condition to provide conservative threshold estimates. Next, we binned the EVS signal (bin widths of 0.05 mA) and computed the mean and variance of the lateral force distribution. Specifically, we used a bin bounded between −0.025 and +0.025 mA for estimating

the lateral force distribution at an EVS of 0 mA, whereas subsequent bins were centred on multiples of ±0.05 mA. The separation between two lateral force distributions was calculated using *d'* measure [eqn. (1)] from signal detection theory (Green & Swets, 1966):

$$d'(\text{EVS}) = \frac{|\mu(\text{EVS}) - \mu(0)|}{\sqrt{(\sigma^2(\text{EVS}) + \sigma^2(0))/2}} \qquad (1)$$

where $\mu(\text{EVS})$ and $\sigma^2(\text{EVS})$ represent the mean and variance of the lateral force distribution at a given EVS magnitude and $\mu(0)$ and $\sigma^2(0)$ represent the mean and variance of the lateral force distribution when the EVS magnitude was 0 mA. The numerator of eqn (1) is proportional to the gain of the response (i.e. gain is the slope of the lateral force time series as a function of the EVS time series), whereas the denominator of eqn (1) corresponds to the variability in the response (i.e.

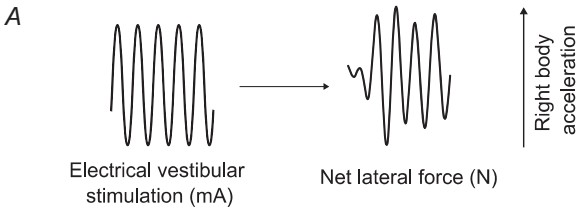

*A*

Electrical vestibular stimulation (mA) Net lateral force (N) Right body acceleration

*B*

Current amplitude 0.4 mA

**Figure 1. Calculation of vestibular-evoked balance control threshold**
Examples of EVS (electrical vestibular stimulation) and bandpass-filtered lateral force time series (*A*). The plot of the lateral force time series as a function of the EVS time series (*B*, upper panel) and lateral force distribution with the mean and standard deviation with and without EVS. The overlap between the two lateral force distributions was measured by *d'*, which varied linearly with EVS signal. The two filled circles illustrate the predicted means and variances for EVS amplitudes of 0 and −0.26 mA (i.e. where *d'* = 1). Note that for these analyses the means and variances were calculated at bins centred around multiples of ±0.05 mA (see Methods). Linear fit (*B*, lower panel) to interpolate between data points where means and variances were calculated to determine the detection thresholds. The vestibular-evoked balance control thresholds were computed as the EVS amplitude at which *d'* = 1.

distribution of the lateral force time series at given EVS magnitudes).

As *d'* increased linearly as a function of EVS amplitude (similar to Jamali et al., 2014; Kwan et al., 2019; Sadeghi et al., 2007), we used a linear fit to interpolate between data points to determine the detection threshold (Fig. 1*B*). When the calculated EVS thresholds (i.e. *d'* = 1) were greater than the peak EVS amplitude, we extrapolated the fit lines to estimate these threshold values. Then we considered only thresholds from trials with a coefficient of determination ($r^2$) > 0.8. For the 0.2 mA EVS condition at 0.1, 0.2, 0.5 and 1 Hz, 80.0%, 97.1%, 100.0% and 100.0% of the trials were included in calculating the vestibular-evoked balance control threshold. For the 0.4 mA EVS condition, the percentages of trial retained were 70.0%, 92.9%, 100.0% and 100.0%, whereas for the 0.6 mA EVS condition, the percentages were 70.0%, 97.1%, 100.0% and 100.0%. The vestibular-evoked balance control threshold was computed as the minimum of the absolute value of the positive and negative values of the EVS for which *d'* equals 1 (e.g. Sadeghi et al., 2007). To assess how spurious correlations between the EVS and force signals affected the calculation of vestibular-evoked balance control thresholds, we simulated EVS signals (i.e. sham EVS signal) with amplitudes of 0.2, 0.4 and 0.6 mA at frequencies of 0.1, 0.2, 0.5 and 1 Hz and used the force data from the trial without EVS. Then, we replicated the same analysis steps for all sham EVS amplitude and frequency combinations as for the EVS conditions. For the EVS conditions, we also calculated the gain of the relationship between the lateral force and the EVS signal and variability in its distribution at three EVS amplitudes (+0.2, 0, −0.2 mA; the lower EVS amplitude condition determined this choice). We calculated the gain of the relationship using the difference in means of the lateral force distribution divided by the difference in applied EVS where we estimated these force distributions. The variability in the lateral force distributions was calculated using the denominator of eqn (1). We performed these analyses to determine if the changes in threshold associated with EVS frequency and amplitude were due to the changes in response gain or variability.

When balancing upright, the lateral force (i.e. CoM acceleration) contains energy between the 0 and ∼1 Hz frequencies (Fitzpatrick et al., 1996; Latt et al., 2003; Loram et al., 2005). To confirm that the EVS induced lateral force oscillations at the target frequency above the sensorimotor noise associated with upright balance, we computed the amplitude spectral density of the bandpass-filtered lateral force signals for EVS and sham EVS conditions by taking the absolute value of the function *fft* in MATLAB (version 2022b, MathWorks, Natick, MA, USA). We expected larger-amplitude spectral peaks for the EVS conditions, as vestibular stimuli evoking a balance response will induce lateral force oscillations at the stimulated frequency. We quantified these changes by computing and comparing the peak spectral amplitude at the stimulus frequency between the EVS and sham EVS conditions.

Finally, we related balance thresholds evoked by EVS (measured in mA) to the predicted signals of virtual head movement (in °/s and m/s²). This transformation allowed us to discuss and compare the identified vestibular-evoked balance thresholds with perceptual vestibular thresholds previously reported in the literature, particularly those reported during head tilt rotation in seated participants (Allred & Clark, 2023). To achieve this, we used a mechanistic model of vestibular processing (Khosravi-Hashemi et al., 2019) that combines equations relating how EVS and mechanical head stimuli modulate primary vestibular afferents and central processing of vestibular self-motion signals. This allowed us to predict the virtual signals of head angular velocity and linear acceleration evoked by the various EVS conditions. We performed all simulations using MATLAB (version 2022b, MathWorks) with the code provided here: https://doi.org/10.5683/SP2/KPMTOH. As inputs to the model, we used the median (and 25th and 75th percentiles) vestibular-evoked balance control thresholds (in mA) for each participant across different EVS amplitudes and frequencies to predict the equivalent head kinematics (angular velocity and inferred interaural linear acceleration).

## Statistical analysis

First, we verified if participants perceived the motion induced by low-amplitude EVS. Perceptual data above the perceptual threshold for a one-interval detection task (i.e. 70.7%, the threshold for a 2-down/1-up staircase paradigm; see Merfeld, 2011) were considered perceived. For the remaining nine EVS conditions (see Table 1), we first computed the ratio of detected abnormal whole-body motion and the number of trials in the condition of interest. We then compared this ratio between each EVS condition and sham EVS using non-parametric permutation testing using MATLAB (version 2022b, MathWorks). We randomly shuffled condition labels (EVS and sham) 10,000 times to generate an empirical distribution of *t* test values based on chance. We then compared this distribution to the observed count differences between conditions. To calculate the *p*-value, we used *z*-scores (the difference between EVS and sham conditions divided by the standard deviation of the empirical distribution) and set the significance level at 0.0056 (0.05/9 comparisons). We also used non-parametric permutation testing (i.e. pre-planned *t* test, as we expected more energy during

**Table 1. Perception of whole-body motion as a function of vestibular stimuli amplitude and frequency (mean ± SD and range).**

|         | 0.2 mA | 0.4 mA | 0.6 mA |
|---------|--------|--------|--------|
| 0.1 Hz  | 25% ± 23% [7–71] | 43% ± 27% [14–93] | *66% ± 18% [43–100] |
| 0.2 Hz  | 30% ± 18% [14.29–71.40] | 59% ± 26% [29–100] | *68% ± 26% [29–100] |
| 0.5 Hz  | 38% ± 33% [0–100] | *75% ± 35%* [33–100] | *75% ± 35%* [0–100] |
| 1.0 Hz  | *63% ± 23% [50–100] | 63% ± 44% [0–100] | *88% ± 35%* [0–100] |
| Sham EVS | | 25% ± 20% [0–53] | |

EVS conditions) to compare peaks of the amplitude spectral density at the target frequency for EVS and sham conditions.

The main statistical analyses were performed using R (R Core Team, 2023) and the ARTool package (Wobbrock et al., 2011). Before statistical analyses were performed, we assessed the normality of distributions using Shapiro–Wilk's test. Because the threshold, the gain and the variability data were not normally distributed, we performed aligned rank transform (ART) tests using ARTool. ARTool performs non-parametric factorial ANOVA. This procedure consists of a preliminary step of data alignment based on the mean estimates of the main and interaction effects of a given factorial model followed by a rank assignment. To analyse the thresholds, gains and variability as a function of EVS amplitude and frequency, we performed separate two-way repeated-measures ANOVA on each transformed variable, with EVS amplitude and frequency as factors [3 amplitudes (0.2, 0.4, 0.6 mA) × 4 frequencies (0.1, 0.2, 0.5 and 1 Hz)]. Vestibular-evoked balance control thresholds represent the minimum of the absolute value of the positive and negative values of the EVS, for which $d'$ equals 1. Although we chose to report the absolute vestibular-evoked balance control threshold, as it is the minimal threshold for a given condition, we verified whether thresholds differed for the anode right–cathode left (inducing right body acceleration) and anode left–cathode right (inducing left body acceleration) electrical vestibular stimuli. We performed a Wilcoxon signed-rank test for each experimental condition to assess symmetry, because the data were not normally distributed. We set the statistical significance level at $p < 0.05$ and performed this analysis using JASP (JASP Team 2024, version 0.18.3). As the threshold, gain and variability data were not normally distributed, we reported these values as median and interquartile (i.e. 25th and 75th percentiles).

## Results

During the experiment no participant reported cutaneous sensation behind their ears. We first determined if the low-amplitude (0.2–0.6 mA) sinusoidal EVS evoked perceptions of motion. We asked participants ($n = 8$), after each trial (i.e. EVS and sham conditions), if they felt their whole-body motions were larger than their natural whole-body motions. Participants reported larger-than-natural whole-body motions for trials with and without EVS (sham EVS). The frequency of these reports increased with the increasing amplitude and frequency of EVS (Table 1). Across all EVS frequencies, participants reported on average (±SD) larger-than-natural whole-body motions in 39% ± 6%, 60% ± 13% and 74% ± 10% of the trials for EVS amplitudes of 0.2, 0.4 and 0.6 mA, respectively. For trials without EVS (sham EVS), participants reported larger-than natural postural sways for 25% ± 20% of the trials. Participants reported abnormal balance motion at a rate lower than the 70.7% perceptual threshold (numbers in bold, Table 1) that also did not differ from the perceptual rate observed in the sham EVS conditions for the 0.2 mA EVS at 0.1 Hz ($p = 0.50$), 0.2 Hz ($p = 0.23$) and 0.5 Hz ($p = 0.20$), as well as for the 0.4 mA EVS at 0.1 Hz ($p = 0.09$), 0.2 Hz ($p = 0.01$) and 1 Hz ($p = 0.02$). For the other conditions with perception data smaller than the 70.7% perceptual threshold (i.e. 0.2 mA at 1.0 Hz, 0.6 mA at 0.1 Hz and 0.6 mA at 0.2 Hz), participants felt abnormal whole-body motion more often than sham condition ($p = 0.0054$, $p = 0.002$ and $p = 0.0037$, respectively).

Numbers in italics refer to perception data larger than the perceptual threshold for a one-interval detection task (i.e., 70.7%) whereas numbers in bold represent perception data smaller than the 70.7% perceptual threshold. The asterisks indicate EVS conditions when participants reported abnormal whole-body motion more frequently than in the sham EVS condition, only for

conditions with fewer reports than the 70.7% perceptual threshold.

When low-amplitude sinusoidal EVS was applied at all frequencies (0.1, 0.2, 0.5 and 1 Hz), we observed well-characterized changes in lateral forces along the frontal plane at the frequency of the EVS signals (Fig. 2, second row, EVS amplitude of 0.4 mA illustrated). We observed similar vestibular-evoked lateral forces for all participants for the three EVS amplitudes (0.2, 0.4 and 0.6 mA). We then confirmed that the applied low-amplitude EVS evoked balance responses at the target stimulation frequency by characterizing the peak amplitude spectral density in the lateral force signals. As expected, we observed a peak in the lateral force signals at the frequency of the applied EVS signal (Fig. 2, third row). Although we also observed energy at these frequencies in the sham EVS trials because the oscillations associated with whole-body motion while balancing, these were small, and the force signals were variable with respect to the EVS signals. Consequently the peaks in the lateral force spectral amplitude were 66% (range: 24–97), 236% (range: 91–321), 312% (range: 258–356) and 166% (range: 156–188) greater across EVS amplitudes than sham EVS for the 0.1, 0.2, 0.5 and 1 Hz stimuli,

respectively. Compared to sham EVS, these larger peaks in spectral amplitude were significant for EVS frequencies >0.1 Hz (EVS 0.2 mA: $p = 0.04$, $p = 0.009$ and $p = 0.02$; EVS 0.4 mA: $p = 0.009$, $p = 0.004$, $p = 0.006$; EVS 0.6 mA: $p = 0.005$, $p = 0.006$, $p = 0.02$) but not for the 0.1 Hz EVS frequency ($p = 0.25$, $p = 0.20$ and $p = 0.17$ for EVS of 0.2, 0.4 and 0.6 mA).

To quantify vestibular-evoked balance control thresholds, we determined the relationship between the EVS and lateral force signals by plotting the time series of the lateral force as a function of the EVS signal (Fig. 3, first row). We first examined the effects of the current magnitude on the balance threshold to vestibular stimuli. For the 0.2 mA EVS condition, we observed a steeper slope (i.e. gain) in the lateral force *versus* EVS current magnitude than for the 0.4 and 0.6 mA conditions. For the illustrated 0.2 Hz data, the calculated EVS current at $d' = 1$ was smaller than the peak EVS amplitude. For the 0.2 mA EVS condition, however, we observed vestibular-evoked balance control thresholds >0.2 mA for 6 out of 10 participants (range: 0.22–0.33 mA) at 0.1 Hz, 3 out of 10 participants (range: 0.21–0.37 mA) at 0.2 Hz, 3 out of 10 participants (range: 0.21–0.45 mA) for 0.5 Hz and 2 out of 10 participants (0.55 and 0.85 mA)

## EVS amplitude 0.4 mA

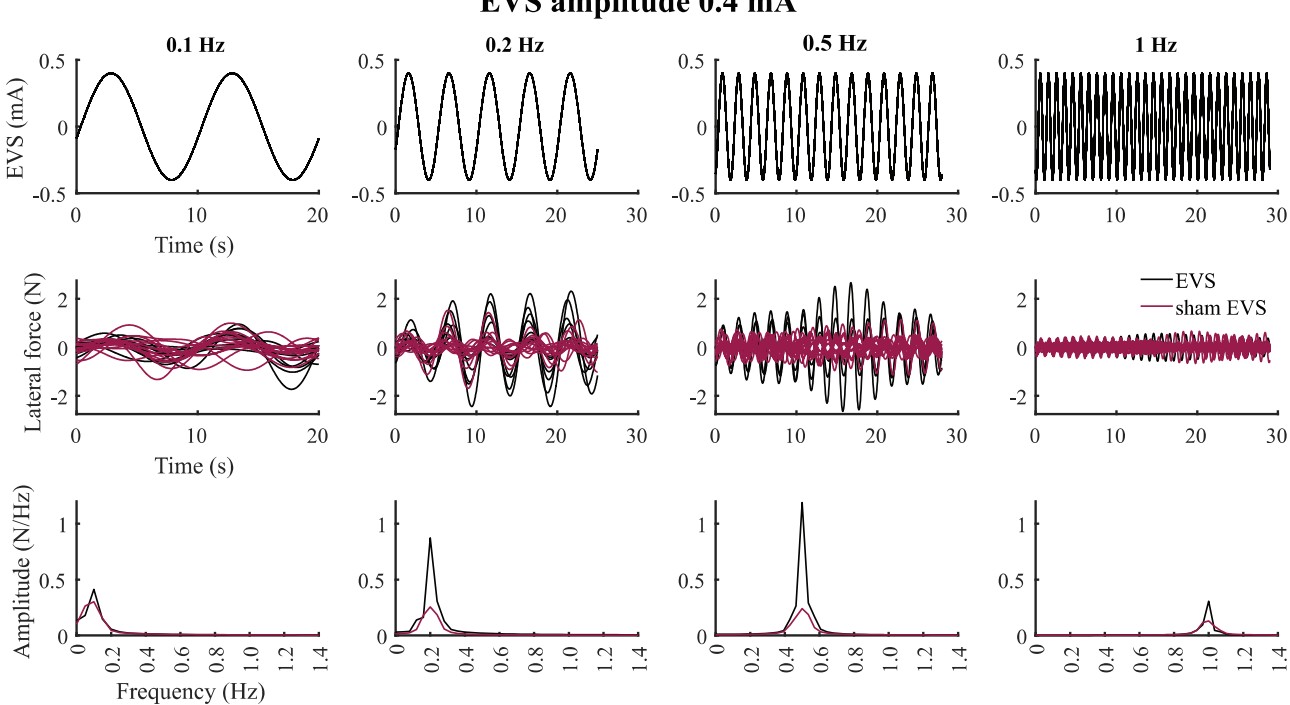

**Figure 2. Characteristics of the vestibular-evoked lateral force to sham and sinusoidal vestibular stimuli**
First row: EVS (electrical vestibular stimulation) time series during sham EVS and EVS conditions. The time axis differs across frequencies as the first cycle was removed from the analysis (see Methods). Second row: lateral forces at the frequency of the sham EVS signal (purple lines) and at the frequency of the EVS signal (black lines). Third row: amplitude spectral density of the lateral force during sham EVS (purple lines) and EVS (black lines) conditions. Representative data from one participant for conditions without and with sinusoidal vestibular stimuli of amplitude 0.4 mA and frequencies of 0.1, 0.2, 0.5 and 1 Hz. [Colour figure can be viewed at wileyonlinelibrary.com]

for 1 Hz. For the 0.4 mA EVS condition, the slope of the lateral force *versus* EVS current amplitude was also steeper than for the 0.6 mA EVS condition. As a result the balance control threshold (i.e. calculated EVS amplitude at $d' = 1$) increased as a function of EVS amplitude, leading to a larger threshold for the 0.6 mA than the 0.4 and 0.2 mA conditions (Fig. 3, second row).

Confirming the results from the lower energy in force signal observed in the sham EVS trials, we observed low gain (i.e. slope) of the lateral force as a function of the sham EVS. Also the variability in the associated force responses was wide (Fig. 4, first row) due to the variability in the relative phase between signals, resulting in high balance control thresholds estimated in the absence of EVS (Fig. 4, second row). These observations were supported by the 1.2–6.9 times (range across EVS amplitude and frequencies) greater vestibular-evoked balance control thresholds estimated in the absence of EVS (i.e. sham EVS) compared to the EVS conditions. These results confirm the physiological origin of the balance thresholds to vestibular stimuli measured with low-amplitude EVS.

Results from all participants (Fig. 5, first row) confirmed that the vestibular-evoked balance control thresholds increased with the amplitude of the EVS [main effect of amplitude: $F_{(2,99)} = 32.28$, $p < 0.001$]. As EVS stimulus amplitude increased from 0.2 to 0.6 mA, the median balance thresholds across participants also increased from 0.23, 0.17, 0.09, 0.10 mA to 0.37, 0.32, 0.27,

0.16 mA and 0.57, 0.37, 0.32, 0.36 mA for EVS frequencies of 0.1, 0.2, 0.5 and 1 Hz, respectively. The calculated thresholds were significantly lower at 0.2 mA compared to both 0.4 and 0.6 mA (multiple $p$-values < 0.001), and the thresholds at 0.4 mA were also significantly lower than that at 0.6 mA ($p = 0.010$). However, the balance thresholds to vestibular stimuli were not influenced by the changes in EVS amplitudes across frequencies [i.e. no amplitude × frequency interaction: $F_{(6,99)} = 0.54$, $p = 0.78$]. We compared the gains and variability of the lateral force distributions to determine why the balance thresholds were affected by EVS amplitude. As for the thresholds, we did not observe an interaction in the gain or variability of the EVS-evoked responses between the EVS amplitude and frequency [gain: $F_{(6,99)} = 1.80$, $p = 0.11$; variability: $F_{(6,99)} = 0.23$, $p = 0.96$]. The gain of the lateral force (Fig. 5, second row, left panel) decreased as a function of EVS amplitude [main effect of amplitude: $F_{(2,99)} = 18.39$, $p < 0.001$]. EVS amplitude, however, did not influence lateral force variability [no main effect of amplitude: $F_{(2,99)} = 0.61$, $p = 0.55$]. These results show that when delivered at a higher amplitude (i.e. 0.6 *vs.* 0.2 mA), EVS induced smaller changes in evoked balance responses for a given change in current.

Consistent with previous reports for vestibular perceptual thresholds to roll mechanical stimuli (Allred & Clark, 2023; Lim et al., 2017; Wagner et al., 2022), our balance control threshold to vestibular stimuli decreased

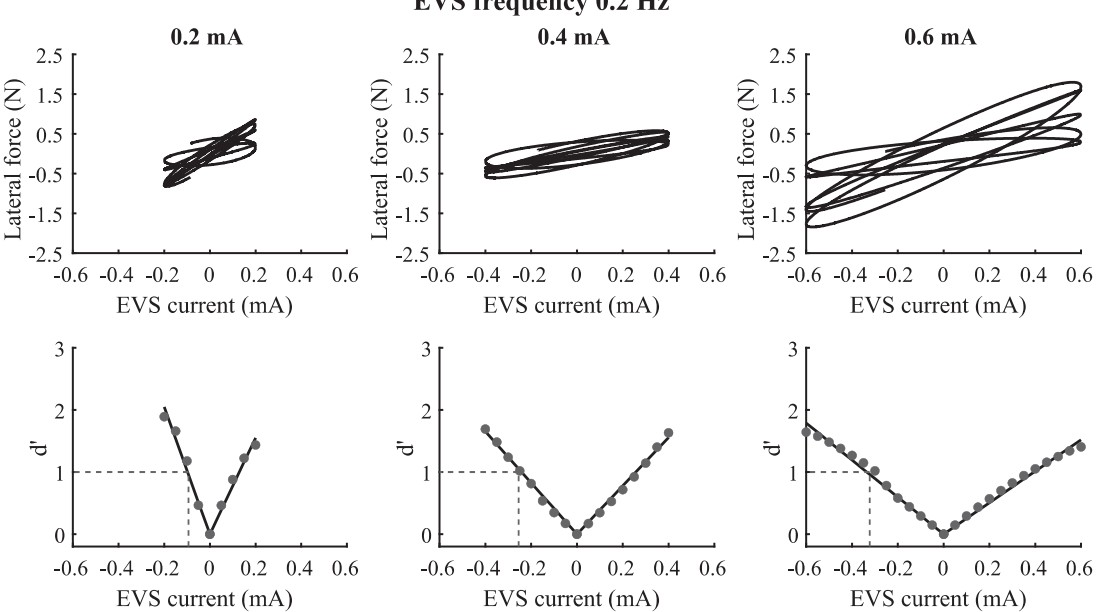

**Figure 3. Balance control threshold to vestibular stimuli**
First row: plot of the lateral force as a function of EVS (electrical vestibular stimulation) time series for EVS amplitude of 0.2 mA (left panel), 0.4 mA (middle panel) and 0.6 mA (right panel) during sinusoidal EVS signal at a frequency of 0.2 Hz. Second row: plot of $d'$ as a function of EVS amplitude. The vestibular balance control threshold was computed as the lowest absolute value of the EVS signal for which $d' = 1$ (dotted horizontal lines). Data from one trial for EVS amplitude of 0.2 mA (left column), 0.4 mA (middle column) and 0.6 mA (right column) are provided.

as a function of EVS frequencies [Fig. 5, first row; main effect of frequency: F(3,99) = 7.15, $p < 0.001$]. Compared to the EVS frequency of 0.1 Hz, the balance threshold was smaller for EVS frequencies of 0.5 and 1 Hz (i.e. $p = 0.011$ and $p < 0.001$) but not different for 0.2 Hz ($p = 0.45$). As both the response gain and variability of the force distribution contribute to the thresholds, we then analysed how each factor was affected by changes in EVS frequency (Fig. 5, second row). Although EVS frequency influenced the gain of the lateral force [F(3,99) = 13.58, $p < 0.001$] and the variability in the force distribution [F(3,99) = 37.15, $p < 0.001$], only the variability in the force exhibited a clear decrease as the EVS frequency increased and contributed to the better separation between distributions (variability was lower for all larger frequencies, multiple $p$-values < 0.001; except for the 0.2 and 0.5 Hz comparison: $p = 0.43$). Indeed the gain of the responses increased from 0.1 to 0.2 Hz ($p = 0.01$), remained constant between 0.2 and 0.5 Hz ($p = 0.28$) and decreased between 0.5 and 1 Hz ($p < 0.001$). This inverted U pattern was distinct from the observed decreases in balance thresholds with frequency. Therefore, vestibular-evoked balance control thresholds decreased as a function of frequency due to reduced variability in the balance responses.

As a secondary analysis, we assessed the symmetry of the estimated balance thresholds. We observed symmetric vestibular balance control thresholds across all EVS amplitudes for EVS frequencies >0.1 Hz (median-range right-left difference in thresholds: 0.0003–0.16 mA; Wilcoxon signed-rank test, multiple $p$-values > 0.05). For the 0.1 Hz stimuli, however, the threshold to elicit left body motion was smaller by 0.14, 0.38 and 0.29 mA at all EVS amplitudes (Wilcoxon signed-rank test, $z = 2.84$, $p < 0.001$, $z = 2.66$, $p < 0.001$ and z = 49.00, $p = 0.03$, for 0.2, 0.4 and 0.6 mA, respectively).

To relate the reported vestibular-evoked balance control thresholds to the predicted virtual head motion signals during upright standing, we computed the equivalent head angular velocity and linear acceleration expected from the computed vestibular-evoked balance control thresholds. We used a mechanistic model of vestibular processing developed by Khosravi-Hashemi et al. (2019) to extract the predicted peak in angular and linear head kinematics. The model predicted that the equivalent head angular velocity (Fig. 6, left panel) and linear acceleration (Fig. 6, right panel) would increase as EVS amplitude increased and decrease as EVS frequency increased, mainly for EVS amplitudes of 0.2 and 0.4 mA. The predicted median head angular velocities from 0.1 to 1 Hz ranged from 0.58 to 0.23, 0.92 to 0.37 and 1.43 to 0.77°/s for EVS amplitudes of 0.2, 0.4 and 0.6 mA, respectively. Correspondingly the predicted interaural median accelerations across frequencies ranged from 0.05 to 0.001, 0.07 to 0.01 and 0.12 to 0.03 m/s² for EVS amplitudes of 0.2, 0.4 and 0.6 mA, respectively.

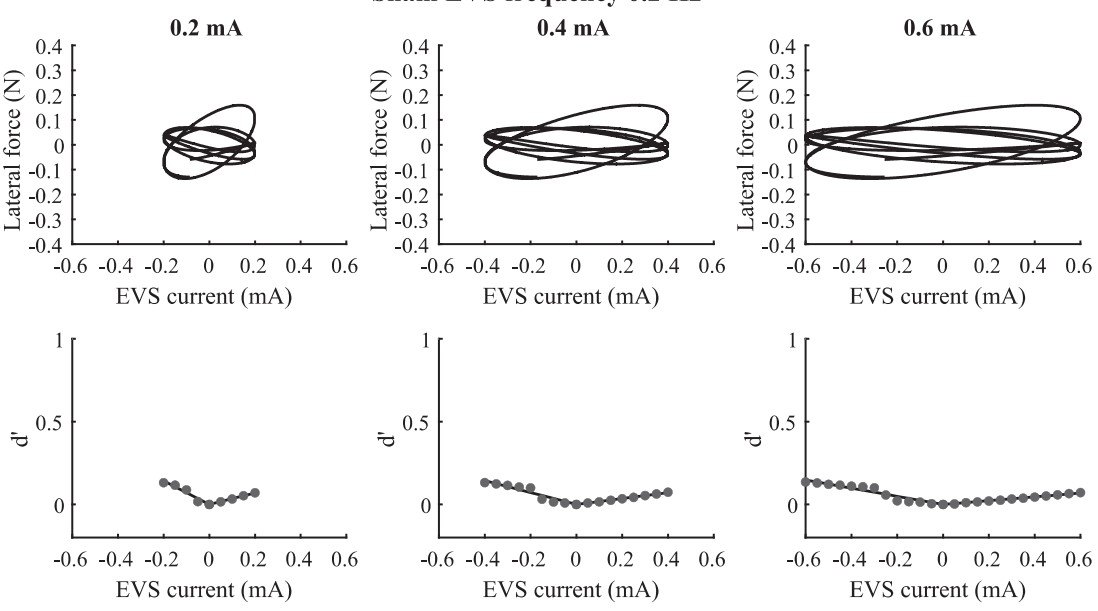

**Figure 4. Vestibular-evoked balance control threshold to sham vestibular stimuli**
First row: plot of the lateral force as a function of the sham EVS (electrical vestibular stimulation) time series for EVS amplitude of 0.2 mA (left panel), 0.4 mA (middle panel) and 0.6 mA (right panel) during sham EVS signal at the 0.2 Hz frequency. Second row: plot of $d'$ as a function of the sham EVS amplitude. The vestibular-evoked balance control threshold, $d' = 1$, is not shown. Data for one trial are shown. Data in the first row are similar because the lateral force is the same; only the sham EVS signal changed.

## Discussion

As spontaneous whole-body oscillations while balancing reflect sensorimotor uncertainty (Kiemel et al., 2006; Peterka, 2002; Simoneau & Teasdale, 2015; van der Kooij & Peterka, 2011; Viseux et al., 2024), we characterized balance control thresholds to evaluate sensorimotor noise within the vestibular control of standing. Using signal detection theory, we computed balance thresholds to isolated electrical vestibular stimuli that varied in amplitude (0.2–0.6 mA) and frequency (0.1–1 Hz). For the 0.2 mA EVS currents, participants reported larger-than-typical whole-body motions in less than 40% of the trials (except for 1 Hz). However, this probability increased with higher stimulus amplitude and frequency. In line with our hypotheses, balance thresholds were lower for the 0.2 and 0.4 mA stimulus amplitudes than 0.6 mA. The larger balance thresholds to 0.6 mA vestibular stimuli resulted from reduced changes in evoked balance responses for a given change in current amplitude (i.e. lower gain). Supporting our second hypothesis, we also observed a reduction in the vestibular-evoked balance control thresholds as the frequency of the vestibular stimuli increased. These frequency-dependent changes

in balance control thresholds were associated with reduced variability in the balance responses (lateral force). Altogether, our results suggest that the signal-to-noise ratio within the vestibular control of balance improved for vestibular stimuli delivered at smaller amplitudes and higher frequencies that remained mostly unperceived.

### Balance threshold to vestibular electrical stimuli

In the current study, we computed balance control thresholds to vestibular electrical stimuli without relying on participants reporting the direction or amplitude of their whole-body motion. First, we determined if participants perceived abnormal body motion associated with low-amplitude EVS (0.2–0.6 mA). A potential issue with perception is that participants may report a sensation in the absence of an actual stimulus. We quantified these false positives using sham trials (no EVS trials), during which participants reported atypical body motion in up to 53% of the trials (mean: 25%). Participants reported atypical whole-body motions in ~25%–38% of trials for the 0.2 mA EVS between 0.1 and 0.5 Hz (not different from sham EVS). For an EVS amplitude of 0.4 mA and

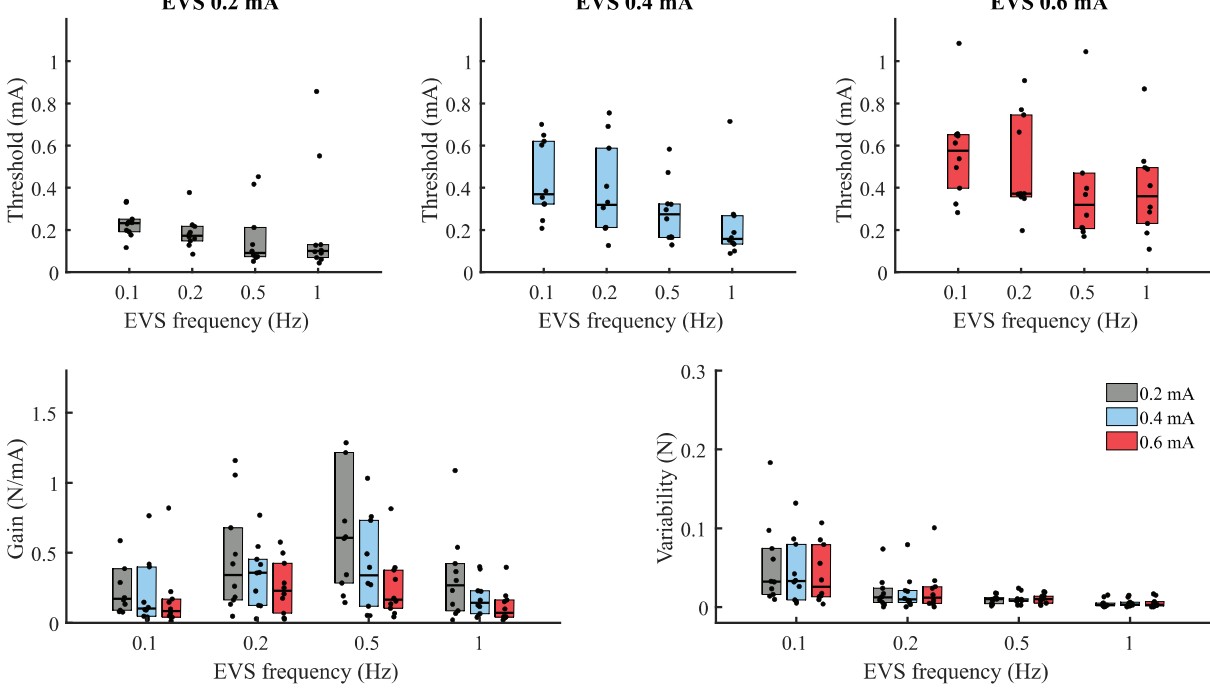

**Figure 5. Threshold and response gain and variability to vestibular stimuli**
First row: vestibular-evoked balance control threshold as a function of EVS (electrical vestibular stimulation) frequency and amplitude. Median-group thresholds for EVS amplitude of 0.2, 0.4 and 0.6 mA for the frequency at 0.1, 0.2, 0.5 and 1 Hz. Second row: median-group gain (left panel) and variability (right panel) of the lateral force across EVS amplitude and frequency. On each panel, black dots represent the mean for each participant, and the black horizontal lines indicate group medians (*n* = 10). The bottom and top edges of the boxes represent the group's 25th and 75th percentiles. One participant's gain at an EVS amplitude of 0.2 mA and a frequency of 0.1 and 0.5 Hz were 3.1 and 2.0 N/mA (second row, left panel). These means are not shown. [Colour figure can be viewed at wileyonlinelibrary.com]

a frequency of 0.1 and 0.2 Hz, this percentage increased to ∼43% and 59% of the trials (not different from sham EVS). Because these perceptual reports of abnormal whole-body balance motion were smaller than the 70.7% perceptual threshold for a one-interval detection task and not different from trials without EVS, we suggest the balance thresholds identified for these stimuli are mostly free of perceptual interference. Next, we showed that EVS induced an ∼67%–313% (range across all amplitude conditions) increase in the lateral ground reaction forces compared to the no-EVS (sham) trials. These results further established that low-amplitude EVS evoked whole-body balance responses despite leading to absent or occasional sensations of body oscillations.

Using a quantitative method (i.e. signal detection theory), we then computed vestibular-evoked balance control thresholds ranging from 0.09 to 0.57 mA. We confirmed that our vestibular-evoked balance control thresholds were not simply due to spurious correlations between the EVS and lateral force signals: (1) EVS evoked consistent lateral force responses, and (2) low-amplitude EVS yielded much smaller (<50%) balance control thresholds than a sham no-EVS condition. Our thresholds matched or were lower than values (0.2–0.5 mA) from a previous study estimating the balance control threshold using a 3-s vestibular electrical square wave pulse (Bent et al., 2000). To determine these thresholds, the authors increased EVS amplitude in 0.05 mA increments until body sways were 'definitely' observed, and participants reported feeling disoriented. The thresholds Bent et al. (2000) reported are also within the range of other studies using similar procedures (Hlavacka et al., 1999;

Inglis et al., 1995). In a more recent study, Mikhail et al. (2021) assessed the EVS-evoked threshold of upright-standing participants using a statistical threshold (head acceleration exceeding 1 SD). The authors determined vestibular thresholds between 1.35 (subjective movement perception by the participant) and 1.94 mA (head acceleration >+1 SD of baseline head acceleration). Differences in methods used to estimate vestibular threshold (e.g. head acceleration, 1 SD) may partially explain discrepancies with the thresholds reported here, but the absence of statistical difference between the thresholds they identified using subjective and statistical methods indicates that the authors' approach could not estimate vestibular thresholds below perception. Indeed, Ertl et al. (2018) reported vestibular perceptual thresholds to 0.5–2 s step pulse EVS in upright-sitting or supine participants between 1.76 and 1.90 mA. Although we lack a definitive explanation for the variability in standing participants' thresholds between our estimates (0.09–0.57 mA) and those reported previously (ranging from 0.2 to 1.94 mA; Bent et al., 2000 and Mikhail et al., 2021), we propose that our methods reflect thresholds of the vestibular control of balance by relying on signal statistics and signal detection theory that avoid subjective assessments of whole-body motion.

### Effect of EVS amplitude on vestibular-evoked balance control thresholds

We delivered three EVS amplitudes to evaluate how increasing the vestibular signals of (virtual) head motion would influence the balance control thresholds. We

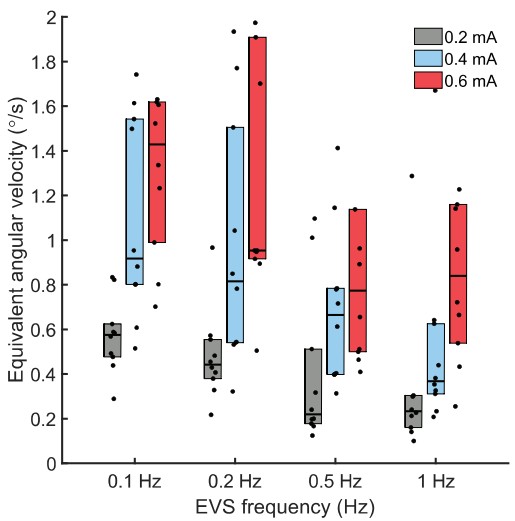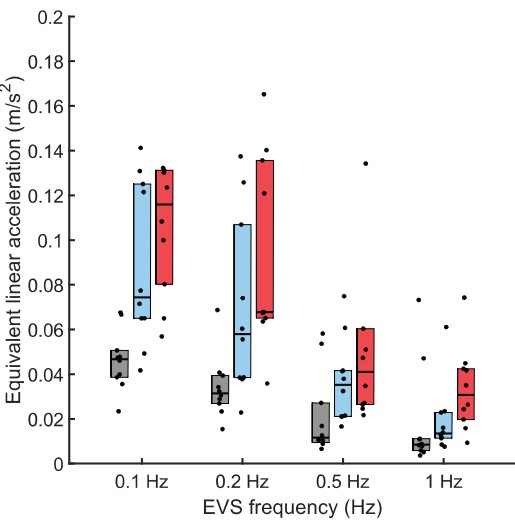

**Figure 6. Model predictions of virtual head kinematics using computed thresholds**
Equivalent head angular velocity (left panel) and linear head acceleration (right panel) across EVS (electrical vestibular stimulation) amplitude and frequency. On each panel black dots represent the mean for each participant, and the black horizontal lines indicate group medians (*n* = 10). The bottom and top edges of the boxes represent the group's 25th and 75th percentiles. [Colour figure can be viewed at wileyonlinelibrary.com]

observed that the balance thresholds to vestibular stimuli increased with stimulus amplitude. Correspondingly these amplitude-dependent balance control thresholds were associated with decreased response gain with larger stimuli, but the variability in balance-evoked responses remained consistent across different EVS amplitudes. Therefore the brain increased the gain of the vestibulomotor balance networks while responding to small vestibular perturbations (i.e. small EVS amplitude). Increasing EVS amplitude could induce sensory reweighting (i.e. decrease vestibular weight) due to the larger variability in head motion signals induced by EVS and the effects of unnatural modulation of the vestibular afferents on the canal–otolith integration mechanisms (Cenciarini & Peterka, 2006; Peterka, 2002). The increased CoM acceleration signals associated with larger EVS amplitude (below and above perception) were more likely to be detected by non-vestibular feedback that could attenuate the EVS-evoked response and contribute to the vestibular gain decreases with larger EVS amplitudes. A related consequence may be that the reduced gain in the vestibular control of balance minimizes EVS-induced instability at higher current amplitudes, as previously reported (Peterka, 2012). The current study, however, was not designed to test the sensory reweighting hypothesis because we did not alter visual or proprioceptive information. Also, the stable vestibular contribution to balance induced by noisy EVS an order of magnitude larger than the currents used here (see non-coherent vestibular stimulus in Héroux et al., 2015), and the inference of linear accelerations to EVS showing functioning canal–otolith integration mechanisms (Khosravi-Hashemi et al. 2019) suggests that another mechanism should be considered. We propose that the observed gain adjustments were performed to detect imposed body motions from sensorimotor noise that resulted in maintaining the signal-to-noise ratio within a desired range. Indeed, due to the similar variability in the force oscillations observed across EVS amplitude conditions at the stimulated frequency, a larger gain in the vestibular control of balance is needed to sense and generate a balance-correcting response to small, imposed perturbations (i.e. EVS amplitudes). As the error signal increased (i.e. larger EVS amplitude), the balance control could decrease its sensorimotor gains while appropriately sensing and responding to perturbations given the similar noise. Response gains that vary as a function of stimulus amplitude suggest the presence of non-linear elements in the vestibular networks controlling standing balance. Due to the presence of non-linearities in the encoding of EVS by primary vestibular afferents (Forbes et al., 2023) and in the encoding/central processing of vestibular cues of large head motion (Carriot et al., 2021; Massot et al., 2012), we suggest that similar processes explain the current sensori-

motor responses. Future studies, however, are needed to identify the source of the non-linearities in the vestibular control of balance (Day et al., 2010).

An important consideration related to increases in EVS amplitude is the potential perception of the vestibular-evoked balance responses. For example vestibular-evoked balance control thresholds were lower for 0.4 than for 0.6 mA EVS. However, participants consistently reported atypical whole-body motions more frequently for the larger EVS amplitude (66%–88% *vs*. 43%–75%). Because force response variability did not change across EVS amplitudes, it is unlikely that interfering perceptual detection/response mechanisms added noise within the sensorimotor loops and increased the balance control threshold to vestibular stimuli. Consequently, the current results do not support the possibility that perceptual motion detection processes render the whole feedback balance control system more variable, at least for the EVS amplitudes tested here.

### Effect of EVS frequencies on vestibular-evoked balance control thresholds

As hypothesized, we observed that vestibular-evoked balance control thresholds decreased as a function of EVS frequency. We have further shown that the decrease in balance control thresholds (0.23 to 0.10, 0.36 to 0.15 and 0.57 to 0.35 mA, for EVS amplitude of 0.2, 0.4 and 0.6 mA) as EVS frequency increased from 0.1 to 1 Hz reflects a reduction in the lateral force variability, improving the signal-to-noise ratio. A higher signal-to-noise ratio for higher vestibular frequency also relates to the low-frequency dynamics of standing balance. Indeed natural CoM accelerations (proportional to lateral forces) associated with standing balance decrease as frequency increases (see Fig. 2; sham EVS amplitude spectral density), resulting in smaller vestibular stimuli needed to evoke balance responses exceeding the natural variability in lateral forces. The frequency-dependent reduction in balance threshold is consistent with changes in vestibular afferent thresholds for EVS frequencies from 0.1 to 1 Hz (Kwan et al., 2019) and vestibular perceptual roll-tilt thresholds (Allred & Clark, 2023; Lim et al., 2017; Wagner et al., 2022) decreasing from 4.11 to 0.86°/s for 0.1 and 0.5 Hz frequencies, respectively. To compare our EVS results with perceptual vestibular thresholds assessed using physical stimuli, we transformed our computed balance thresholds (estimated in mA) to equivalent head angular velocities and linear accelerations using a mechanistic model of vestibular processing (Khosravi-Hashemi et al., 2019). The model confirmed balance thresholds to equivalent head angular velocity and linear acceleration decreased as a function of EVS frequency (Fig. 6). For 0.1 Hz vestibular stimuli

balance thresholds to equivalent head angular velocity were about 4.2 times lower than the reported vestibular perceptual thresholds (0.96 *vs*. 4.11°/s), and for 0.5 Hz stimuli, they were about 1.5 times lower than the previously reported roll perceptual thresholds (0.55 *vs*. 0.86°/s). For the 0.5 Hz vestibular stimuli, the equivalent head angular velocities for EVS amplitudes of 0.4 mA (0.66°/s) and 0.6 mA (0.77°/s) were nearly identical to perceptual thresholds. This difference also corresponds to participants reporting an abnormal balance motion with a lower probability for the 0.5 Hz EVS at 0.2 mA than 0.4 and 0.6 mA (38% *vs*. 75% of trials). Consequently, we propose that the non-perceptual balance threshold to vestibular stimuli-evoked balance can be assessed with EVS amplitudes between 0.2 and 0.4 mA at 0.1 and 0.2 Hz. However when 0.5 Hz EVS is used, the signal amplitude should be limited to 0.2 mA. This suggestion is supported by participants reporting abnormal whole-body motion as frequently for these EVS conditions than for the sham (no EVS) condition.

### Recommendations, applications and limitations

Altogether, the results of the current study showed that 0.1–0.5 Hz EVS with a 0.2 mA amplitude and 0.1–0.2 Hz EVS with a 0.4 mA amplitude permit the quantification of non-perceptual balance control threshold to vestibular stimuli. Because the balance thresholds from six and three participants were larger than the applied 0.2 mA EVS at the 0.1 and 0.2 Hz frequencies, we recommend using 0.2 mA stimuli at 0.5 Hz to assess balance thresholds to vestibular stimuli in healthy controls. This recommendation is also grounded in a better separation of balance-evoked responses from the ongoing balance control at frequencies ≥0.5 Hz. EVS applied at higher frequencies and amplitudes may involve a combination of perceptual and non-perceptual motion detection processes. One way to verify this suggestion will be to measure electrocortical brain activity during upright standing (Fabre et al., 2021) in response to EVS at different amplitudes and frequencies. Our approach could also be expanded to assess balance control thresholds to other sensory stimuli (e.g. proprioceptive and visual stimuli), opening the door to revealing the mechanisms underlying the sensorimotor control of bipedal stance. In addition, the current approach can be used to evaluate the vestibular contributions to balance control in people with sensorimotor impairments. However, careful estimation of the appropriate stimulus parameters needs to be established.

Our research has some limitations. The present recommendations for testing balance control thresholds apply only to healthy young adults; future experiments are needed to expand these recommendations to participants with balance deficits. Sensing and reporting whole-body motions during upright standing is challenging. Although we could have asked participants to report their perception of motion across time using a potentiometer, we chose not to use such a method as hand movements and/or the focus on perceiving and potentially being entrained by the phase of EVS could have altered the lateral force needed to extract balance thresholds. By matching the number of EVS cycles across frequencies, we evaluated participants' perception with fewer trials for higher stimulation frequencies, consequently impacting the calculation of the reported percentage for atypical whole-body motions. Nonetheless, we are confident that the reported percentages for atypical whole-body motions permitted us to estimate participants' perceptions as they did not perceive abnormal motion in ~75% of trials without EVS.

### Conclusion

Using a method based on signal statistics and signal detection theory, we quantified the vestibular-evoked balance control thresholds without relying on subjective evaluation of overall body motions. Although non-perceptual and perceptual detection mechanisms can interact to generate balance-correcting responses to whole-body motion, our results indicate that 0.2 mA EVS stimuli at 0.5 Hz effectively characterize non-perceptual balance thresholds to vestibular stimuli. As the amplitude of vestibular signals increased, the gain in the vestibular control of balance decreased, leading to larger balance thresholds. Balance thresholds decreased as the frequency of the vestibular stimulus increased, likely because of decreased variability in lateral force. These results pave the way for uncovering the physiological mechanisms that regulate the non-perceptual balance-correcting motor commands essential for bipedalism. They will also enable the assessment of individual vestibular contributions to balance control in people with balance impairments and the exploration of multisensory mechanisms that govern balance and overall movement control.

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

## Additional information

### Data availability statement

The data supporting the current study's findings are available from the corresponding author upon reasonable request.

## Competing interests

The authors declare that they have no competing interests.

## Author contributions

M.S.: conceptualization, data curation, software, formal analysis, validation, investigation, visualization, methodology, writing of the original draft, project administration, supervision, funding acquisition, writing – review and editing; M.N.: formal analysis, validation, writing of the original draft, writing – review and editing; J.-S.B.: conceptualization, resources, software, formal analysis, supervision, funding acquisition, validation, investigation, visualization, methodology, writing of the original draft, project administration, writing – review and editing. All authors have approved the final version of the manuscript. All authors agreed to be accountable for all aspects of the work. All persons designated as authors qualify for authorship, and all those who qualify for authorship are listed.

## Funding

This work was funded by the Natural Sciences and Engineering Research Council Discovery Grant RGPIN-2020-06210 (to M.S.) and RGPIN-2020-05438 (to J.-S.B.). M.N. was partly supported by a postdoctoral scholarship from the Centre for Interdisciplinary Research in Rehabilitation and Social Integration (Cirris).

## Acknowledgements

The authors thank all the participants involved in this study and Jesse M. Charlton for his help with the statistical model.

## Keywords

standing balance, non-perceptual balance threshold, sensori-motor noise, vestibular

## Supporting information

Additional supporting information can be found online in the Supporting Information section at the end of the HTML view of the article. Supporting information files available:

**Peer Review History**

