## [Peer Review History · The Journal of Physiology]

Balance control threshold to vestibular stimuli

Martin Simoneau, Mujda Nooristani, and Jean-Sébastien Blouin

DOI: 10.1113/JP288016

Corresponding author(s): Martin Simoneau (Martin.Simoneau@kin.ulaval.ca)

The following individual(s) involved in review of this submission have agreed to reveal their identity: Callum J Osler (Referee #2)

Review Timeline:

Submission Date:	30-Oct-2024
Editorial Decision:	22-Nov-2024
Revision Received:	24-Jan-2025
Editorial Decision:	17-Feb-2025
Revision Received:	25-Feb-2025
Accepted:	12-Mar-2025

Senior Editor: Richard Carson

Reviewing Editor: Ross Pollock

Transaction Report:

Dear Dr Simoneau,

Re: JP-RP-2024-288016 "Balance control threshold to vestibular stimuli" by Martin Simoneau, Mujda Nooristani, and Jean-Sébastien Blouin

Thank you for submitting your manuscript to The Journal of Physiology. It has been assessed by a Reviewing Editor and by 2 expert referees and we are pleased to tell you that it is acceptable for publication following satisfactory revision.

REVISION CHECKLIST:

We look forward to receiving your revised submission.

Yours sincerely,

Richard Carson
Senior Editor
The Journal of Physiology

REQUIRED ITEMS

- Author photo and profile. First or joint first authors are asked to provide a short biography (no more than 100 words for one author or 150 words in total for joint first authors) and a portrait photograph. These should be uploaded and clearly labelled together in a Word document with the revised version of the manuscript. See Information for Authors for further details.
- Your manuscript must include a complete Additional Information section, including competing interests; funding; author contributions and acknowledgements.
- The Journal of Physiology funds authors of provisionally accepted papers to use the premium BioRender site to create high resolution schematic figures. Follow this link and enter your details and the manuscript number to create and download figures. Upload these as the figure files for your revised submission. If you choose not to take up this offer, we require figures to be of similar quality and resolution. If you are opting out of this service to authors, state this in the Comments section on the Detailed Information page of the submission form. The link provided should only be used for the purposes of this submission. Authors will be charged for figures created on this premium BioRender account if they are not related to this manuscript submission.
- Please upload separate high-quality figure files via the submission form.
- Papers must comply with the Statistics Policy: https://jp.msubmit.net/cgi-bin/main.plex?form_type=display_requirements#statistics.

In summary:

- If $n \leq 30$, all data points must be plotted in the figure in a way that reveals their range and distribution. A bar graph with data points overlaid, a box and whisker plot or a violin plot (preferably with data points included) are acceptable formats.
- If $n > 30$, then the entire raw dataset must be made available either as supporting information, or hosted on a not-for-profit repository, e.g. FigShare, with access details provided in the manuscript.
- 'n' clearly defined (e.g. x cells from y slices in z animals) in the Methods. Authors should be mindful of pseudoreplication.

- All relevant 'n' values must be clearly stated in the main text, figures and tables.

- The most appropriate summary statistic (e.g. mean or median and standard deviation) must be used. Standard Error of the Mean (SEM) alone is not permitted.

- Exact p values must be stated. Authors must not use 'greater than' or 'less than'. Exact p values must be stated to three significant figures even when 'no statistical significance' is claimed.

- Please include an Abstract Figure file, as well as the Figure Legend text within the main article file. The Abstract Figure is a piece of artwork designed to give readers an immediate understanding of the research and should summarise the main conclusions. If possible, the image should be easily 'readable' from left to right or top to bottom. It should show the physiological relevance of the manuscript so readers can assess the importance and content of its findings. Abstract Figures should not merely recapitulate other figures in the manuscript. Please try to keep the diagram as simple as possible and without superfluous information that may distract from the main conclusion(s). Abstract Figures must be provided by authors no later than the revised manuscript stage and should be uploaded as a separate file during online submission labelled as File Type 'Abstract Figure'. Please also ensure that you include the figure legend in the main article file. All Abstract Figures should be created using BioRender. Authors should use The Journal's premium BioRender account to export high-resolution images. Details on how to use and access the premium account are included as part of this email.

Reviewing Editor:

Methods Details:

A reference number for the ethical approval should be given.

Comments to ensure the paper complies with the Statistics Policy:

Please ensure that specific p-values are given for all statistical tests. A number of statements in the results should have p-values associated with them, for example line 357 indicates that participants reported larger than natural sway - there should be p-values to go with this, this applies to a number of other areas in the results.

When reporting participant characteristics at the beginning of the methods section please ensure that you indicate that it, presumably, mean (SD).

Comments to the authors:

Thank you for submitting your manuscript to the Journal of Physiology. Having been reviewed by two expert reviewers some points have been raised for you to consider with the manuscript. Many of the point relate to improving the clarity of what is being said but please also consider the 1st major point noted from reviewer one about the interpretation of some of the statements.

Please ensure you include a reference number for the ethical approval that was obtained.

Senior Editor:

Comments to ensure the paper complies with the Statistics Policy:

To reiterate comments provided by the Referees and Reviewing Editor, please ensure not only that exact p values are given, but also (if not already the case) that the relevant test statistic is given along with the corresponding degrees of freedom. If additional tables are required, that is of course acceptable.

In relation to instances in which the absence of a difference is being inferred (and indeed more generally) , please also consider providing effect size estimates along with corresponding confidence intervals.

Referee #1:

General comments.

The study characterizes human balance responses using very low amplitude sinusoidal currents to electrically evoke vestibular responses (EVS responses). Results are presented indicating that the test subjects did not perceive balance disturbances to the low amplitude EVS but that corrective balance responses were robustly present. This result clearly demonstrates that balance control mechanisms are operating at levels below conscious perception. This is an important contribution since it highlights the differences between motion perception, which has been used to investigate sensory processing and motor control, and the underlying processes that influence motor control but are not perceived. Additionally, the methods are novel and could be applied to investigate the contributions of other sensory systems to balance and motor control. The study is well designed and used appropriate analysis techniques. A weakness is described in my first major comment where I believe that an alternative interpretation needs to be considered and I request some clarification on two additional points.

Major comments:

1. Page 5 lines 119-125. The end of the paragraph can be interpreted to imply that sensory sources other than vestibular only contribute to balance control when they are perceived. This is similarly implied in the sentence beginning on line 132 and this sentence also hypothesizes that higher vestibular EVS thresholds would occur at higher EVS amplitudes that produce larger balance disturbances that are perceived by other sensory systems. This reviewer does not see any reason to think that other sensory systems (in this case somatosensation/proprioception in eyes closed conditions) are not contributing to balance control in the region of very small balance disturbances evoked by the low amplitude EVS. Evidence for this comes from the Peterka 2002 paper that found a dominant (70%) contribution of proprioceptive cues when subjects were tested eyes closed using very low amplitude surface-tilt stimuli that evoked body sway that was only slightly greater than spontaneous sway. This result was supported by results in Cenciarini & Peterka (J Neurophysiology 2006) that used combined EVS and surface-tilt stimuli to investigate vestibular and proprioceptive contributions to balance. To make the argument that non-vestibular sensory sources only contribute when balance disturbances are perceived, one would have to argue that if Peterka 2002 had used even lower amplitude surface-tilt stimuli there would be some very low stimulus amplitude where the proprioception contribution would drop to zero and subjects would then be relying only on vestibular information. Much more likely is that balance control investigated using very low amplitude perturbations of any kind is still investigating a multisensory integration and control problem. When this is recognized, then a potential alternative explanation for the reduction in EVS gain and increasing statistically-defined threshold with increasing EVS amplitude (Figure 5) is that this could be due to a sensory reweighting phenomenon where a larger EVS amplitude is adding more

variability to the vestibular signals and the CNS is compensating by down-weighting the vestibular contribution to balance and upweighting use of proprioceptive signals. It is worth noting that EVS affects the variability of otolith afferents and therefore the quality/precision of central canal-otolith integration. Thus, even though the EVS effect is a net roll motion vector derived from the summation of canal responses (I think you're assuming this), the EVS effect on otolith afferents could also contribute to a less precise (noisier) central estimate of head movement and orientation that could evoke sensory reweighting perhaps to a greater extent than natural head rotation of the same magnitude. Finally, the 'equivalent angular velocities' shown in Figure 7 might allow for some very rough comparisons to the results in Peterka 2002 (although that study evoked AP sway). For 0.1 Hz, 0.2 mA EVS Figure 7 shows about 0.7deg/s equivalent angular velocity (peak). This corresponds to ~1.1 deg peak angular displacement which is an RMS amplitude of ~0.8 deg. RMS body sway amplitudes from Peterka 2002 in the eyes-closed surface-tilt condition were below this 0.8 deg value and this amount of sway was well within the range where sensory re-weighting was evident.

2. Page 19, line 465. I don't understand what is being said in the sentence beginning with 'Hence'. Usually when there is a 'hence' the sentence is drawing a conclusion that connects two or more things that were previously stated. Here, there are the statements that (1) lateral force gain decreased with increasing EVS amplitude and (2) EVS amplitude did not influence lateral force variability. How does this last 'hence' sentence connect these two points?

3. Page 19, last sentence beginning on line 483. I don't understand this last sentence. Your equation for threshold (line 241) says that for a given response amplitude the threshold should be larger when the variability is lower but this sentence seems to say the opposite. Also, the sentences earlier in the paragraph state that there was not a consistent decrease in thresholds with frequency (sentence beginning on line 417).

Minor Comments:

1. Page 8, line 217. I don't understand the numbers that are given in the parentheses. Just previously it is mentioned that median thresholds differences were less than 0.01 mA. Therefore, I would expect the numbers in the parentheses to be listing these small differences.

2. Figure 1 legend. I think additional description is needed in the legend to indicate that the middle plot is showing two example points where means and variances were calculated with one of those points (at about -0.18 mA) corresponding to the current level that gave a d' value of 1.0.

3. Page 14, line 356. Suggest 'Across all EVS frequencies . . .'

4. Page 14, line 361 and Table 1. The 'black numbers' do not look much blacker than other numbers. Maybe bolding is better.

5. Page 15, Figure 2. The Methods say that trials were 40 s. Why are these plots all less than 40 s and also are not the same duration for the different frequencies?

6. Page 16, line 406. Suggest 'For the illustrated 0.2 Hz data, . . .'

7. Page 16 beginning at line 406. This is mentioning that some d' values were greater than the peak EVS amplitude. Presumably you extrapolated the fit lines to get these values. This should be mentioned in the Methods section when describing how d' was calculated.

8. Page 16, lines 412, 413. Clarify that the first two numbers in parentheses are for 0.5 Hz data and the last two are for 1.0 Hz data (I'm guessing that's true).

9. Page 20, line 513. Should be 'angular and linear'.

10. Page 20, Figure 5. Suggest that it would be useful to use the same color code in the top row of plots as was used in the bottom row. So the bars in the top middle plot would all be blue and the bars in the top right plot would all be red.

Referee #2:

Simoneau and colleagues have used signal detection theory in the context of the vestibular control of balance to develop an original method of establishing balance control thresholds. Amplitude- and frequency-dependent changes in threshold are shown, along with data to ascertain the underlying physiological mechanisms. A pre-existing model has then been used to estimate corresponding values in terms of angular and linear motion for comparison with existing literature. I found the paper enjoyable to read and data are reported very nicely using a series of figures. The study is well designed; for instance, a sham condition has been used to demonstrate that the established thresholds are not merely an artefact of the analysis. My main question relates to the amount body sway that is in fact induced by the EVS stimuli (as outlined in my comments below). I have also suggested some areas for improvement in terms of stating the research questions within the introduction, along with some minor clarifications throughout. Nonetheless, this interesting paper offers a new method of establishing thresholds and highlights the distinction between perception and balance control in this context.

ABSTRACT

Line 63 - "0.17-0.46 mA" - it is unclear to me how these values relate to the data in the results. For example, thresholds of 0.43, 0.42, 0.33 and 0.42 mA are reported for 0.6 mA, which would presumably result in an average lower than 0.46.

INTRODUCTION

Lines 116-119 - unnecessary repetition of "the EVS-evoked virtual head motion signals result in balance-correcting responses" and "consequently, EVS evokes balance responses"

Lines 132-137 - while some hypotheses/predictions are made concerning the effects of EVS amplitude and the underlying mechanism, it is not clear what evidence/previous research these predictions are based on. (Unlike the predictions on the effects of frequency which are clearly based on previous findings).

Lines 142-149 - this discussion of the results is difficult for the reader to follow at this point in the paper (i.e. before the methods have been described and/or the supporting data have been presented). Overall, the last paragraph of the introduction could be improved by simply stating the questions to be addressed and the associated hypotheses.

METHODS

In accordance with the Journal of Physiology's Information for Authors, the Methods should start with a paragraph headed 'Ethical approval'.

Line 159 - "participants stood upright on two force platforms" - could you please provide stance width and confirm whether participants were barefoot/shoed

Line 171-173 - "To determine if participants felt the motion induced by EVS" - in several other places the current paper uses phrases such as "stimuli below perceptual thresholds" and "unperceived sensory stimuli". But the perceptual measure used appears to be about the response to EVS (i.e. response rather than stimuli). To me, it seems that it would be difficult for the participant to separate perception of the stimulus (virtual) from perception of the response (actual). Could you please clarify exactly what question(s) was(were) asked to participants and whether the stimulus and/or response were explicitly mentioned.

Line 192 - "toward the opposite side" - opposite side has no meaning here, as the direction of the virtual motion signal (i.e. anodal/cathodal) has not been described in prior statements.

Lines 216-218 - this information is difficult to follow at this stage of the paper because a) procedures for calculating threshold and retained trials have not been described yet, b) the corresponding threshold values for 80 cycles are not reported, and c) only the 0.6 mA condition has been reported for retained trials (what about other stimulus amplitudes?). Please consider re-writing and/or relocating in the paper.

Line 236/Figure 1B - "bin widths of 0.05 mA" - could you please clarify the bin widths used as it is unclear to me how these bin widths correspond to the data points on Figure 1B, which do not appear to be at 0.05 mA intervals. Also, was +/-0.025 mA used for the 0 mA value?

Line 266-269 - please clarify how gain and variability were calculated .

Line 312 - "~70%" - is the approximate (i.e. "~") needed here?

RESULTS

In accordance with the Journal of Physiology's Information for Authors, tests of significance should be specified on each occasion and in full.

To fully interpret the perception of whole-body motion data, it would be useful to also present actual whole-body motion data (e.g. using a traditional measure of sway, possibly derived from centre of pressure if body sway not measured). This would allow the reader to understand whether/by how much the participants' actual sway increased in each condition. This relates to my earlier point about whether participants are reporting perception of the virtual stimulus (vestibular) or the actual response (multiple senses). I appreciate that the amplitude spectral density of the lateral ground reaction force has been shown to increase in EVS conditions but - given the body's inertia - it is unclear by how much actual sway of the body is increased.

Figure 2 - is there a reason the time-series graphs end one period short of 30 seconds? (i.e. why not 40 seconds?).

Line 460 - "in the gain..." - remove this phrase, as interaction statistics reported are for gain and variability (i.e. not just gain).

Line 513 - "angular an linear" should read "angular and linear"

DISCUSSION

Line 573 - "0.17-0.46 mA" - same point as point raised above for the abstract

Lines 587-589 - there are also differences in the method used to establish the 1.94 mA threshold (e.g. head acceleration, 1 SD), which could explain the calculated threshold differs to the current study.

Line 592 - "(~3-9 x thresholds)" - the meaning of this is unclear to me. Could you please clarify?

'Effect of EVS amplitude on vestibular-evoked balance control thresholds' section - the effects of EVS amplitude on the response gain/slope (i.e. N/mA) are explained by changes in central vestibular processing. While I tend to agree with this explanation, is it possible to rule out an alternative explanation based on veridical non-vestibular feedback? That is to say, once the participant begins to sway, sensory feedback of this (actual) motion attenuates the response, and this effect is more pronounced at higher stimulus amplitudes where the evoked sway is greater. Relating back to my earlier point, to ascertain whether this is a possibility, it would be interesting to know how much actual body sway is evoked by the stimuli at different amplitudes/frequencies.

END OF COMMENTS

Québec City, January 22nd, 2025

Dear Editor Carson,

We thank you for the opportunity to revise our paper. Please find a revised version of our manuscript entitled “Balance control threshold to vestibular stimuli” that we hope will be suitable for publication in the Journal of Physiology. We thank the Reviewing Editor and Reviewers for their helpful and constructive feedback. Based on their comments, we feel the manuscript has improved in clarity, quality, and impact. We also include a point-by-point response to each comment raised by the Reviewing Editor and Reviewers. Based on the reviewer's comment regarding providing more details on the analyses, we revisited the script and identified a minor inconsistency with previous approaches (e.g., Kwan et al., 2019). We adjusted the analyses accordingly, which resulted in smaller thresholds overall but otherwise did not alter the study's main findings. We have also revised the results and figures accordingly. The modifications within the revised manuscript related to reviewers' comments are denoted by "red text".

Comments to the authors:

Thank you for submitting your manuscript to the Journal of Physiology. Having been reviewed by two expert reviewers some points have been raised for you to consider with the manuscript. Many of the point relate to improving the clarity of what is being said but please also consider the 1st major point noted from reviewer one about the interpretation of some of the statements.

Response: Thank you. As requested, we addressed all comments raised by the reviewers and provided a detailed response regarding the interpretation of our results in response to Reviewer 1. In addition, we also included the interpretation proposed by Reviewer 2 in the revised discussion.

Please ensure you include a reference number for the ethical approval that was obtained.

Response: We added the ethical approval to the Methods in the revised manuscript (H17-02672).

Reviewing Editor:

A reference number for the ethical approval should be given.

Response: We added the ethical approval to the Methods in the revised manuscript (H17-02672).

Comments to ensure the paper complies with the Statistics Policy:

Please ensure that specific p-values are given for all statistical tests. A number of statements in the results should have p-values associated with them, for example line 357 indicates that participants reported larger than natural sway - there should be p-values to go with this, this applies to a number of other areas in the results.

When reporting participant characteristics at the beginning of the methods section please ensure that you indicate that it, presumably, mean (SD).

Response: As requested, we have added the specific p-values for all results from statistical tests, including the perceptual data highlighted by the Editor. We have also clarified that the \pm represents SD. Note that for p-values smaller than 0.001, we report $p < 0.001$ to avoid reporting $p = 0.0000$.

Referee #1:

General comments.

The study characterizes human balance responses using very low amplitude sinusoidal currents to electrically evoke vestibular responses (EVS responses). Results are presented indicating that the test subjects did not perceive balance disturbances to the low amplitude EVS but that corrective balance responses were robustly present. This result clearly demonstrates that balance control mechanisms are operating at levels below conscious perception. This is an important contribution since it highlights the differences between motion perception, which has been used to investigate sensory processing and motor control, and the underlying processes that influence motor control but are not perceived. Additionally, the methods are novel and could be applied to investigate the contributions of other sensory systems to balance and motor control. The study is well designed and used appropriate analysis techniques. A weakness is described in my first major comment where I believe that an alternative interpretation needs to be considered and I request some clarification on two additional points.

Response: We thank the reviewer for evaluating the paper positively and recognizing the importance of measuring balance thresholds outside of perceptual mechanisms. We have modified the paper according to the reviewer's recommendations and offer a point-by-point answer to their comments below.

Major comments:

1. Page 5 lines 119-125. The end of the paragraph can be interpreted to imply that sensory sources other than vestibular only contribute to balance control when they are perceived. This is similarly implied in the sentence beginning on line 132 and this sentence also hypothesizes that higher vestibular EVS thresholds would occur at higher EVS amplitudes that produce larger balance disturbances that are perceived by other sensory systems. This reviewer does not see any reason to think that other sensory systems (in this case somatosensation/proprioception in eyes closed conditions) are not contributing to balance control in the region of very small balance disturbances evoked by the low amplitude EVS. Evidence for this comes from the Peterka 2002 paper that found a dominant (70%) contribution of proprioceptive cues when subjects were tested eyes closed using very low amplitude surface-tilt stimuli that evoked body sway that was only slightly greater than spontaneous sway. This result was supported by results in Cenciarini & Peterka (J Neurophysiology 2006) that used combined EVS and surface-tilt stimuli to investigate vestibular and proprioceptive contributions to balance. To make the argument that non-vestibular sensory sources only contribute when balance disturbances are perceived, one would have to argue that if Peterka 2002 had used even lower amplitude surface-tilt stimuli there would be some

very low stimulus amplitude where the proprioception contribution would drop to zero and subjects would then be relying only on vestibular information. Much more likely is that balance control investigated using very low amplitude perturbations of any kind is still investigating a multisensory integration and control problem. When this is recognized, then a potential alternative explanation for the reduction in EVS gain and increasing statistically-defined threshold with increasing EVS amplitude (Figure 5) is that this could be due to a sensory reweighting phenomenon where a larger EVS amplitude is adding more variability to the vestibular signals and the CNS is compensating by down-weighting the vestibular contribution to balance and upweighting use of proprioceptive signals. It is worth noting that EVS affects the variability of otolith afferents and therefore the quality/precision of central canal-otolith integration. Thus, even though the EVS effect is a net roll motion vector derived from the summation of canal responses (I think you're assuming this), the EVS effect on otolith afferents could also contribute to a less precise (noisier) central estimate of head movement and orientation that could evoke sensory reweighting perhaps to a greater extent than natural head rotation of the same magnitude. Finally, the 'equivalent angular velocities' shown in Figure 7 might allow for some very rough comparisons to the results in Peterka 2002 (although that study evoked AP sway). For 0.1 Hz, 0.2 mA EVS Figure 7 shows about 0.7deg/s equivalent angular velocity (peak). This corresponds to ~1.1 deg peak angular displacement which is an RMS amplitude of ~0.8 deg. RMS body sway amplitudes from Peterka 2002 in the eyes-closed surface-tilt condition were below this 0.8 deg value and this amount of sway was well within the range where sensory re-weighting was evident.

Response: We thank the reviewer for these thoughtful comments and suggestions regarding the interpretation of our results.

First, we agree with the reviewer that there is no reason to think that other sensory systems (in this case somatosensation/proprioception in eyes closed conditions) are not contributing to balance control in the region of very small balance disturbances evoked by the low amplitude EVS. To address this point, we have changed the wording of the paragraph highlighted (and subsequent lines) by the reviewer to avoid potential confusion:

Consequently, when applied above a certain current amplitude, EVS-evoked balance-correcting responses can be perceived (as opposed to the virtual head motion signals; Wardman et al., 2003a). To avoid issues related to the perceptual detection of imposed perturbations, we used signal detection theory (Green & Swets, 1966) to quantify balance responses that remained unperceived but were evoked by vestibular cues of head motion above the sensorimotor noise underlying the control of natural whole-body oscillations while standing upright.

Given that the larger whole-body movements induced by higher EVS currents were more likely to be perceived and potentially induce balance responses of non-vestibular origin, we hypothesized that higher EVS amplitudes would lead to higher balance thresholds to vestibular stimuli.

Second, the reviewer suggests an alternative explanation for the reduction in EVS gain due to sensory reweighting. To address this comment, we have mentioned this possibility in the Discussion. While we recognize the importance of the sensory reweighting hypothesis and that our experiment was not designed to test this hypothesis, we do not believe this is a major

contributing factor to the present observations. First, we did not modify the other sensory contributions to balance to test the weight of the vestibular contributions to balance. The reviewer implies that increasing EVS amplitude could induce a reweighting by itself due to the larger variability induced by EVS. In our experiment, EVS represents a virtual head motion signal at the tested frequencies instead of a noisy (broadband frequency input) signal that could be attenuated to estimate whole-body motion better. This statement is supported by the results from a previous experiment where participants did not use sensory reweighting to minimize the destabilizing effects induced by noisy EVS (see non-coherent vestibular stimulus in Héroux et al. 2015). Instead, our previous observations clearly demonstrated that participants recalibrated EVS-related head motion signals within their internal representation of whole-body motion to enable the control of standing balance in the presence of EVS up to 10 mA. Given the lack of sensory reweighting observed for EVS amplitudes more than 10 times larger than in the current experiment, we argue that although the reweighting hypothesis could explain our observations, previous evidence does not support this statement. We also appreciate and thank the reviewer for suggesting that the canal-otolith integration mechanisms may be affected by the unnatural activation of the vestibular afferents by EVS and contribute to sensory reweighting. We cannot exclude this possibility but note, however, that EVS applied to healthy participants revealed perceptions of linear accelerations that were dependent on head orientation with respect to gravity (Khosravi-Hashemi et al. 2019). Although we recognize that these results cannot exclude that the canal-otolith integration mechanism was unaffected by EVS, they reveal that these mechanisms are still functioning. Here, we proposed that a potential mechanism underlying the control of standing balance is the maintenance of a signal-to-noise ratio to detect imposed body motions from sensorimotor noise, where the EVS represents a virtual signal of head motion instead of a noisy signal that the brain should minimize given the variance associated with this signal.

Regarding the observations from Peterka (2002) cited by the reviewer, we prefer to remain cautious regarding comparisons with our observations. Nevertheless, we provide additional analyses in this response to the Reviewer to support our interpretation. Although we agree that the angular velocity values below which sensory reweighting occurred in Peterka's classic work may be close to the equivalent head motion resulting from EVS we reported, motion-equivalent EVS represents an input virtual signal of head motion, whereas the observations from Peterka relate to the CoM body sway (RMS) in response to the input perturbations. As requested by Reviewer #2, we quantified the RMS value of the COM angular body sway in response to all EVS and sham EVS conditions (see figure below) to better compare our observations to the values reported by Peterka (2002). For the lowest EVS amplitude (i.e., 0.2 mA), the RMS of the COM angular body sway across participants were 0.34°, 0.38°, 0.27° and 0.36°, for EVS frequencies of 0.1, 0.2, 0.5 and 1 Hz compared to 0.25° for sham EVS (p-values: 0.08, 0.06, 1.00, and 0.05 for 0.2 mA @ 0.1, 0.2, 0.5 and 1 Hz, respectively). Overall, the median RMS COM angular body sway remained below 0.5° for all conditions (except for 0.6 mA @ 0.1 Hz), that is, at or below the reported sway angle values by Peterka (2002) in response to support surface tilts up to 2° peak-to-peak. Finally, Peterka (2002) reported that for low amplitude stimuli (up to 2° peak-to-peak), body-sway response initially increased with stimulus, but saturation was observed for larger stimulus amplitudes. This relationship between whole-body motion and the input low

amplitude perturbation stimulus appears distinct from our observations, leading us to propose the alternate interpretation mentioned above.

We have revised the Discussion to add sensory reweighting as a hypothesis and presented it first given its importance in the field: *'Increasing EVS amplitude could induce sensory reweighting (i.e., decrease vestibular weight) due to the larger variability in head motion signals induced by EVS and the effects unnatural activation of the vestibular afferents on the canal-otolith integration mechanisms. However, the absence of sensory reweighting to minimize the destabilising effects induced by noisy EVS an order of magnitude large than the currents used here (see non-coherent vestibular stimulus in Héroux et al. 2015) and the inference of linear accelerations to EVS showing functioning canal-otolith integration mechanisms (Khosravi-Hashemi et al. 2019) suggest another mechanism must be considered'*.

2. Page 19, line 465. I don't understand what is being said in the sentence beginning with 'Hence'. Usually when there is a 'hence' the sentence is drawing a conclusion the connects two or more things that were previously stated. Here, there are the statements that (1) lateral force gain decreased with increasing EVS amplitude and (2) EVS amplitude did not influence lateral force variability. How does this last 'hence' sentence connect these two points?

Response: We thank the reviewer for this comment. We have revised the sentence starting with 'Hence' by: *These results show that when delivered at a higher amplitude (i.e., 0.6 vs 0.2 mA), EVS induced smaller changes in evoked balance responses for a given change in current.*

3. Page 19, last sentence beginning on line 483. I don't understand this last sentence. Your equation for threshold (line 241) says that for a given response amplitude the threshold should be larger when the variability is lower but this sentence seems to say the opposite. Also, the sentences earlier in the paragraph state that there was not a consistent decrease in thresholds with frequency (sentence beginning on line 417).

Response: The equation for quantifying balance thresholds is based on signal-detection theory, where the separation between distributions can be calculated using the means and variances of the compared distributions. Related to the current experiment, a lower variability of the lateral force distributions will lead to a better discriminability between these distributions and consequently reach the target $d'=1$ for lower EVS stimulus amplitudes. To address this comment, we modified Figure 2B to show an arrow on the x-axis representing the balance threshold and revised the sentence highlighted by the reviewer by adding: *'... and contributed to the better separation between distributions...'*

Regarding the consistent decrease in thresholds with frequency, the sentence provides results comparing the thresholds elicited by the 0.2, 0.5, and 1 Hz stimuli to the 0.1 Hz stimulus. It does not describe a specific trend in the data. Our analyses show that the higher stimulus frequencies (0.5 and 1 Hz) yielded lower balance control thresholds than the 0.1 Hz stimulus. Still, the threshold difference was insignificant between the 0.1 and 0.2 Hz stimuli. A note of caution is

required given that we did not observe an amplitude \times frequency interaction (requiring frequency comparisons averaged across all EVS amplitudes) and because stimuli at higher EVS amplitudes included responses that the participants perceived.

Minor Comments:

1. Page 8, line 217. I don't understand the numbers that are given in the parentheses. Just previously it is mentioned that median thresholds differences were less than 0.01 mA. Therefore, I would expect the numbers in the parentheses to be listing these small differences.

Response: We thank the reviewer for this comment. We clarified in the revised manuscript that we performed this analysis given that the number cycles was larger for the 1 Hz EVS frequency. We also simplified the presentation of numbers in bracket. We have revised the text to: *'Thus, the remaining cycles across EVS frequencies for each stimulus amplitude were 28 for 0.1 Hz, 35 for 0.2 Hz, 42 for 0.5 Hz, and 58 for 1 Hz. Given the larger number of cycles for the 1 Hz EVS condition (58 cycles), we also estimated vestibular-evoked balance control thresholds with 42 cycles. These analyses revealed lower median thresholds with fewer cycles (up to 0.01 mA for 42 cycles). Given the minimal differences and the expected decrease in vestibular thresholds with higher EVS frequency, we only presented data from the 58 cycles for the 1 Hz EVS condition to provide conservative threshold estimates.'*

2. Figure 1 legend. I think additional description is needed in the legend to indicate that the middle plot is showing two example points where means and variances were calculated with one of those points (at about -0.18 mA) corresponding to the current level that gave a d' value of 1.0.

Response: We added the following sentence to the Figure 1 legend: *'The two filled circles illustrate the predicted means and variances for EVS amplitudes of 0 and 0.26 mA (i.e., where $d' = 1$). Note that for these analyses, the means and variances were calculated at bins centered around multiples of ± 0.05 mA (see Methods).'* We thank the reviewer for this comment. Note that we depict data from another trial in the revised manuscript, which is more representative of the median threshold for this EVS condition.

3. Page 14, line 356. Suggest 'Across all EVS frequencies . . .'

Response: Edited in the revised manuscript.

4. Page 14, line 361 and Table 1. The 'black numbers' do not look much blacker than other numbers. Maybe bolding is better.

Response: We thank the reviewer for catching this! Instead, we italicized these numbers and revised the caption for Table 1.

5. Page 15, Figure 2. The Methods say that trials were 40 s. Why are these plots all less than 40 s and also are not the same duration for the different frequencies?

Response: We thank the reviewer for this comment. The trial duration was 40s, but EVS lasted 30s with 5s periods without EVS at the beginning and end of the trials to facilitate data alignment. As mentioned in the Method section, we did not include the first cycle when calculating the vestibular-evoked balance control threshold for each EVS frequency condition. This resulted in the time vector duration of 20, 25, 28, and 29 seconds for EVS frequencies of 0.1, 0.2, 0.5, and 1 Hz, leading to different duration illustrated in Figure 2. We have revised the Methods section accordingly: *'Each trial lasted 40 s but included 5 s periods without EVS at the beginning and end of each trial to facilitate data alignment (see below the procedure for aligning both signals).'* Furthermore, we added in the figure legend: *'The time axis differs across frequencies as the first cycle was removed from the analysis (see Method section).'*

6. Page 16, line 406. Suggest 'For the illustrated 0.2 Hz data, . . .'

Response: Edited in the revised manuscript.

7. Page 16 beginning at line 406. This is mentioning that some d' values were greater than the peak EVS amplitude. Presumably you extrapolated the fit lines to get these values. This should be mentioned in the Methods section when describing how d' was calculated.

Response: We revised the Methods section to include: *'When the calculated EVS thresholds (i.e., when $d' = 1$) were greater than the peak EVS amplitude, we extrapolated the fit lines to estimate these threshold values.'*

8. Page 16, lines 412, 413. Clarify that the first two numbers in parentheses are for 0.5 Hz data and the last two are for 1.0 Hz data (I'm guessing that's true).

Response: The reviewer is correct. To address this comment, we added: *'for the 0.5 Hz EVS'* and *'for the 1 Hz EVS'* in the parentheses.

9. Page 20, line 513. Should be 'angular and linear'.

Response: We changed “an” for “and”.

10. Page 20, Figure 5. Suggest that it would be useful to use the same color code in the top row of plots as was used in the bottom row. So the bars in the top middle plot would all be blue and the bars in the top right plot would all be red.

Response: As suggested, we changed the color in the upper panel to match the colors in the lower panel.

Referee #2:

Simoneau and colleagues have used signal detection theory in the context of the vestibular control of balance to develop an original method of establishing balance control thresholds. Amplitude- and frequency-dependent changes in threshold are shown, along with data to ascertain the underlying physiological mechanisms. A pre-existing model has then been used to estimate corresponding values in terms of angular and linear motion for comparison with existing literature. I found the paper enjoyable to read and data are reported very nicely using a series of figures. The study is well designed; for instance, a sham condition has been used to demonstrate that the established thresholds are not merely an artefact of the analysis. My main question relates to the amount body sway that is in fact induced by the EVS stimuli (as outlined in my comments below). I have also suggested some areas for improvement in terms of stating the research questions within the introduction, along with some minor clarifications throughout. Nonetheless, this interesting paper offers a new method of establishing thresholds and highlights the distinction between perception and balance control in this context.

Response: We thank the reviewer for these positive comments and for thoroughly reviewing our paper. We have addressed the concerns below and made the corresponding changes to improve the revised manuscript.

ABSTRACT

Line 63 - "0.17-0.46 mA" - it is unclear to me how these values relate to the data in the results. For example, thresholds of 0.43, 0.42, 0.33 and 0.42 mA are reported for 0.6 mA, which would presumably result in an average lower than 0.46.

Response: The threshold range in the abstract and the Discussion section (i.e., Line 573) was for the mean instead of the median threshold. Using the median values, the threshold range is 0.09-0.57 mA. We have revised the manuscript accordingly and now report the threshold range using the median thresholds.

INTRODUCTION

Lines 116-119 - unnecessary repetition of "the EVS-evoked virtual head motion signals result in balance-correcting responses" and "consequently, EVS evokes balance responses"

Response: We modified the second sentence to: '*Consequently, when applied above a certain current amplitude, EVS-evoked balance-correcting responses...*'.

Lines 132-137 - while some hypotheses/predictions are made concerning the effects of EVS amplitude and the underlying mechanism, it is not clear what evidence/previous research these predictions are based on. (Unlike the predictions on the effects of frequency which are clearly based on previous findings).

Response: We thank the Reviewer for this comment. We have revised the manuscript and added references for work supporting that EVS-evoked responses scale with stimulus amplitude (Latt et al. 2003, Wardman et al. 2003, Day et al. 1997) and that the probability of perceiving a perturbation increases with stimulus amplitude (Teasdale et al. 1999, Tisserand et al. 2022).

Lines 142-149 - this discussion of the results is difficult for the reader to follow at this point in the paper (i.e. before the methods have been described and/or the supporting data have been presented). Overall, the last paragraph of the introduction could be improved by simply stating the questions to be addressed and the associated hypotheses.

Response: To address this comment, we have revised and clarified the presentation of the main results related to the hypotheses. We hope the reviewer will agree that these changes improved readability. *‘However, the mechanism underlying the increase in balance thresholds to larger vestibular stimuli did not support our prediction: larger balance thresholds were associated with a decrease in the gain of the lateral force as stimulus amplitude increased. We further revealed that the reduction in balance thresholds as a function of EVS frequencies was due to a decrease in the variability of lateral force oscillations at higher frequencies.’*

METHODS

In accordance with the Journal of Physiology's Information for Authors, the Methods should start with a paragraph headed 'Ethical approval'.

Response: We added a paragraph headed 'Ethical approval' at the start of the Methods section.

Line 159 - "participants stood upright on two force platforms" - could you please provide stance width and confirm whether participants were barefoot/shoed

Response: The participants stood barefoot on the two force platforms. We normalized the stance width of each participant to their foot length, measured as the distance between the posterior calcaneus and the distal end of the first distal phalanx. We defined stance width as the distance between the base of the left and right 5th metatarsals and added this information to the Procedure section.

Line 171-173 - "To determine if participants felt the motion induced by EVS" - in several other places the current paper uses phrases such as "stimuli below perceptual thresholds" and "unperceived sensory stimuli". But the perceptual measure used appears to be about the response to EVS (i.e. response rather than stimuli). To me, it seems that it would be difficult for the participant to separate perception of the stimulus (virtual) from perception of the response (actual). Could you please clarify exactly what question(s) was(were) asked to participants and whether the stimulus and/or response were explicitly mentioned.

Response: We thank the reviewer for this comment. We want to point the reviewer to the Wardman et al. (2003) paper. In that paper, the authors clearly established that when balancing upright, participants always perceived the direction of the sway as opposed to the direction of the imposed perturbations (they never perceived the direction of the EVS perturbation). This contrasts with when participants were immobilized in an upright position when they could perceive the perturbation's direction after long EVS applications. Our recent data extended this observation to perturbations applied to the whole body while balancing (Tisserand et al., 2022).

Given that our participants balanced freely, they could only perceive the direction of the motion induced by the EVS. We note that we applied cream to eliminate potential cutaneous sensations behind the ears and that the low amplitude of the EVS made it impossible to report a specific direction for the duration of the trials. Instead, participants felt sensations of motions that they could attribute to EVS – although they sometimes reported ‘extra’ motion during the sham trials. We modified the revised manuscript to clarify this point: ‘*When balancing upright, Wardman et al. (2003b) reported that participants always perceived the direction of the sway as opposed to the direction of the imposed perturbations (i.e., they never perceived the direction of the EVS perturbation). Consequently, we determined if participants felt the balance motion induced by EVS by asking them to report if they felt any abnormal whole-body motions after each EVS and sham trial.*’

Line 192 - "toward the opposite side" - opposite side has no meaning here, as the direction of the virtual motion signal (i.e. anodal/cathodal) has not been described in prior statements.

Response: As suggested by the reviewer, we have removed ‘toward the opposite side’ and modified the sentence to: ‘... *which evokes whole-body balance-correcting responses.*’

Lines 216-218 - this information is difficult to follow at this stage of the paper because a) procedures for calculating threshold and retained trials have not been described yet, b) the corresponding threshold values for 80 cycles are not reported, and c) only the 0.6 mA condition has been reported for retained trials (what about other stimulus amplitudes?). Please consider re-writing and/or relocating in the paper.

Response: We thank the reviewer for this comment. We clarified in the revised manuscript that we performed this analysis given that the number cycles was larger for the 1 Hz EVS frequency. We also simplified the presentation of numbers in bracket. We have revised the text to: ‘*Thus, the*

remaining cycles across EVS frequencies for each stimulus amplitude were 28 for 0.1 Hz, 35 for 0.2 Hz, 42 for 0.5 Hz, and 58 for 1 Hz. Given the larger number of cycles for the 1 Hz EVS condition (58 cycles), we also estimated vestibular-evoked balance control thresholds with 42 cycles. These analyses revealed lower median thresholds with fewer cycles (up to 0.01 mA for 42 cycles). Given the minimal differences and the expected decrease in vestibular thresholds with higher EVS frequency, we only presented data from the 58 cycles for the 1 Hz EVS condition to provide conservative threshold estimates.'

Line 236/Figure 1B - "bin widths of 0.05 mA" - could you please clarify the bin widths used as it is unclear to me how these bin widths correspond to the data points on Figure 1B, which do not appear to be at 0.05 mA intervals. Also, was +/-0.025 mA used for the 0 mA value?

Response: Based on the reviewer's comment regarding providing more details on the bin widths, we revisited the script and identified a minor inconsistency with previous approaches (e.g., Kwan et al., 2019). We adjusted the scripts, which resulted in smaller thresholds, but did not alter the main findings of our study. We revised the results and figures accordingly.

To address the reviewer's specific question, the bin was -0.025 to +0.025 mA for EVS at the 0 mA value. Then, the bin widths were 0.025 – 0.075 mA, 0.075 – 0.125 mA, 0.125 – 0.175 mA... for the positive slope. These bins are centered at 0.05, 0.1, 0.15 mA... values. For example, the d' calculated at 0.05 mA includes data from the 0.025 – 0.075 mA bin. For the linear fits, we forced d' to 0 for the 0 mA EVS amplitude (estimated from the -0.025 to +0.025 mA bin).

Line 266-269 - please clarify how gain and variability were calculated.

Response: We added the following text to describe how we calculated gain and variability: '*We calculated the gain of the relationship using the difference in means of the lateral force distribution divided by the difference in applied EVS where we estimate these force distributions. The variability in the lateral force distributions was calculated using the denominator of Eq.1.'*

Line 312 - "~70%" - is the approximate (i.e. "~") needed here?

Response: We initially wrote ~70% because Merfeld (2011) states that the threshold for a one-interval detection for a 2-down/1-up staircase paradigm is 70.7%, and others have used a threshold of 69% for a single interval direction discrimination task (Peters et al. 2015). However, given that we used the Merfeld reference, we changed ~70% to 70.7% in the revised manuscript.

RESULTS

In accordance with the Journal of Physiology's Information for Authors, tests of significance should be specified on each occasion and in full.

Response: We have included the exact p values for each statistical test we report in the paper. Note that for p-values smaller than 0.001, we report $p < 0.001$ to avoid reporting $p = 0.0000$.

To fully interpret the perception of whole-body motion data, it would be useful to also present actual whole-body motion data (e.g. using a traditional measure of sway, possibly derived from centre of pressure if body sway not measured). This would allow the reader to understand whether/by how much the participants' actual sway increased in each condition. This relates to my earlier point about whether participants are reporting perception of the virtual stimulus (vestibular) or the actual response (multiple senses). I appreciate that the amplitude spectral density of the lateral ground reaction force has been shown to increase in EVS conditions but - given the body's inertia - it is unclear by how much actual sway of the body is increased.

Response: We thank the reviewer for this suggestion. As suggested, we estimated the position of the center of mass (COM) along the frontal plane (i.e., the mediolateral axis) by low-pass filtering the position of the center of pressure (Benda et al., 1994, Caron et al., 1997). Here, we used COM to represent the vertical projection of the COM on the ground (also called the center of gravity). The figure below depicts the center of pressure (COP, thick lines) and the COM (thin lines) positions along the frontal plane for EVS amplitude of 0.2, 0.4, and 0.6 mA for a frequency of 0.5 Hz.

We calculated the root mean square (RMS) values of the COM angular sway along the frontal plane with and without vestibular stimuli to represent how participants swayed across conditions (see Figure below). We calculated the COM angular sway by dividing the COM sway along the frontal plane by the height of the COM (i.e., COM vertical position above the ankle joints = $0.575 \times \text{height}$) of the participant. Then, we computed the RMS value. We used a two-sided Wilcoxon signed rank test (between each EVS condition and sham EVS) to verify whether the RMS COM angular sway increased in the presence of EVS compared to sham EVS. For the EVS condition at 0.2 mA, the RMS values of the COM angular sway were 0.34° , 0.38° , 0.27° , and 0.36° for EVS frequencies of 0.1, 0.2, 0.5, and 1 Hz compared to 0.25° for sham EVS. For EVS amplitude of 0.2 mA, we did not observe differences in the RMS value of the COM angular sway

between EVS and sham EVS ($p = 0.08$, $p = 0.07$, $p = 1.00$, and $p = 0.05$, for frequencies of 0.1, 0.2, 0.5, and 1 Hz). The RMS values of the COM angular sway were larger for the 0.4 mA EVS at 0.1 Hz ($p = 0.004$) and 0.2 Hz ($p = 0.01$), but did not differ from sham EVS for frequencies of 0.5 and 1 Hz ($p = 0.06$ and $p = 1.00$, respectively). For the 0.6 mA EVS amplitude, the RMS values of the COM angular sway were larger than sham EVS for the 0.1 Hz ($p = 0.002$), 0.2 Hz ($p = 0.03$), and 1 Hz ($p = 0.002$) frequencies, but no difference was observed the 0.5 Hz frequency ($p = 0.38$). Given that RMS COM angular sway did not show differences between many EVS conditions and sham EVS, we prefer to keep the spectral analyses in the paper due to their better sensitivity.

Figure 2 - is there a reason the time-series graphs end one period short of 30 seconds? (i.e. why not 40 seconds?).

Response: The trial duration was the 40s, but EVS lasted 30s with 5s periods without EVS at the beginning and end of the trials to facilitate data alignment; we clarified that in the revised manuscript. As mentioned in the Methods section, we did not include the first cycle when calculating the vestibular-evoked balance control threshold for each EVS frequency condition. Thus, the time vector duration was 20, 25, 28, and 29 seconds for EVS frequencies of 0.1, 0.2, 0.5, and 1 Hz in Figure 2. We revised the Methods section to include this information. In addition, we stated in the figure legend: *'The time axis differs across frequencies as the first cycle was removed from the analysis (see Method section).'*

Line 460 - "in the gain..." - remove this phrase, as interaction statistics reported are for gain and variability (i.e. not just gain).

Response: We thank the reviewer for identifying this. To represent the data, we modified the sentence to: ‘... we did not observe an interaction in the gain or variability of the EVS-evoked responses between the EVS amplitude and frequency...’.

Line 513 - "angular an linear" should read "angular and linear"

Response: Thank you, we corrected this typo.

DISCUSSION

Line 573 - "0.17-0.46 mA" - same point as point raised above for the abstract

Response: The threshold range in the abstract and the Discussion section (i.e., Line 573) was for the mean threshold. We now report the threshold range using the median thresholds, i.e., 0.09-0.57 mA.

Lines 587-589 - there are also differences in the method used to establish the 1.94 mA threshold (e.g. head acceleration, 1 SD), which could explain the calculated threshold differs to the current study.

Response: We agree with the reviewer. To emphasize the differences in methods between studies, we modified the text to: ‘*Differences in methods used to estimate vestibular thresholds (e.g., head acceleration, 1 SD) may partially explain discrepancies with the thresholds reported here, but the absence of statistical difference between the thresholds they identified with subjective and statistical methods indicates that the authors’ approach could not estimate vestibular thresholds below perception.*’.

Line 592 - "(~3-9 x thresholds)" - the meaning of this is unclear to me. Could you please clarify?

Response: The text was meant to compare vestibular thresholds reported between studies. We clarified the text between parentheses: ‘*(thresholds varied by a factor 3-9 between studies)*’.

'Effect of EVS amplitude on vestibular-evoked balance control thresholds' section - the effects of EVS amplitude on the response gain/slope (i.e. N/mA) are explained by changes in central vestibular processing. While I tend to agree with this explanation, is it possible to rule out an alternative explanation based on veridical non-vestibular feedback? That is to say, once the participant begins to sway, sensory feedback of this (actual) motion attenuates the response, and this effect is more pronounced at higher stimulus amplitudes where the evoked sway is greater. Relating back to my earlier point, to ascertain whether this is a possibility, it would be interesting to know how much actual body sway is evoked by the stimuli at different amplitudes/frequencies.

Response: We thank the reviewer for this comment. As proposed, we did perform analyses of COM movements induced by EVS. Overall, these analyses revealed that EVS up to 0.6 mA increased RMS COM angular sway ($0.27 - 0.70^\circ$) compared to sham EVS (0.25°), but for many EVS conditions COM movements were not significantly larger than for the sham EVS condition (see statistical results above). As proposed by the reviewer, it is possible that veridical non-vestibular feedback could attenuate the response and explain the gain decreases with large EVS amplitudes. We have added this suggestion to the revised Discussion. Please note that we also included the possibility that sensory reweighting plays a role, as proposed by Reviewer #1. We added the following text to the Discussion: ‘*Another factor possibly contributing could be the increased COM acceleration signals associated with larger EVS amplitude. As COM acceleration increased, they would likely be detected by non-vestibular feedback that could attenuate the EVS-evoked response and explain the vestibular gain decreases with larger EVS amplitudes.*’

REFERENCES

- Benda, B.J., P.O. Riley, and D.E. Krebs. 1994. Biomechanical relationship between center of gravity and center of pressure during standing. *Rehabilitation Engineering, IEEE Transactions* [see also *IEEE Transactions on Neural Systems and Rehabilitation*] 2: 3–10.
- Caron, O., B. Faure, and Y. Breniere. 1997. Estimating the centre of gravity of the body on the basis of the centre of pressure in standing posture. *Journal of Biomechanics* 30: 1169–71.
- Day, B. L., Severac Cauquil, A., Bartolomei, L., Pastor, M. A., & Lyon, I. (1997). Human body-segment tilts induced by galvanic stimulation: a vestibularly driven balance protection mechanism. *The Journal of Physiology*, 500(3), 661-672.
- Khosravi-Hashemi, N., Forbes, P. A., Dakin, C. J., & Blouin, J. S. (2019). Virtual signals of head rotation induce gravity-dependent inferences of linear acceleration. *The Journal of Physiology*, 597(21), 5231-5246.
- Latt, L. D., Sparto, P. J., Furman, J. M., & Redfern, M. S. (2003). The steady-state postural response to continuous sinusoidal galvanic vestibular stimulation. *Gait & Posture*, 18(2), 64-72.
- Peterka, R. J. (2002). Sensorimotor integration in human postural control. *Journal of Neurophysiology*, 88(3), 1097-1118.
- Peters, R. M., Rasman, B. G., Inglis, J. T., & Blouin, J. S. (2015). Gain and phase of perceived virtual rotation evoked by electrical vestibular stimuli. *Journal of Neurophysiology*, 114(1), 264-273.
- Tisserand, R., Rasman, B. G., Omerovic, N., Peters, R. M., Forbes, P. A., & Blouin, J. S. (2022). Unperceived motor actions of the balance system interfere with the causal attribution of self-motion. *PNAS nexus*, 1(4), pgac174.

Wardman, D. L., Taylor, J. L., & Fitzpatrick, R. C. (2003). Effects of galvanic vestibular stimulation on human posture and perception while standing. *The Journal of Physiology*, 551(3), 1033-1042.

Dear Dr Simoneau,

Re: JP-RP-2025-288016R1 "Balance control threshold to vestibular stimuli" by Martin Simoneau, Mujda Nooristani, and Jean-Sébastien Blouin

Thank you for submitting your manuscript to The Journal of Physiology. It has been assessed by a Reviewing Editor and by 2 expert referees and we are pleased to tell you that it is acceptable for publication following satisfactory revision.

REVISION CHECKLIST:

We look forward to receiving your revised submission.

Yours sincerely,

Richard Carson
Senior Editor
The Journal of Physiology

EDITOR COMMENTS

Reviewing Editor:

Thank you for considering and responding to the points raised by the reviewers and the changes made have improved the overall quality of the manuscript. You will see that reviewer 1 has raised some points related to the revision and their previous comments. Could you respond to the points being raised and modify the manuscript appropriately or provide the rationale for keeping things as they are.

Also, currently you do not have an acknowledgments section in the manuscript - if appropriate could you include one.

REFEREE COMMENTS

Referee #1:

Comments:

This reviewer was generally satisfied with the revisions but there remain a few items that need to be clarified. I continue to agree that this study represents impactful research for the field, the study design and data analysis was very good, and the results were clearly presented.

However, the response to Reviewer #1's first major comment was not entirely satisfying. Specifically, the issue is about whether non-vestibular sensory systems necessarily have to be perceived before they are able to modify the response to EVS. The authors' explanatory response to the comment states an agreement with the reviewer that ". . . there is no reason to think that other sensory systems . . . are not contributing to balance control in the region of very small balance disturbances evoked by low amplitude EVS." But the modified text (beginning on line 132; "Given that . . .") still implies that it would be necessary for body motion be perceived before non-vestibular sources influence balance responses. This reviewer agrees that larger EVFs that induce body motion large enough to be perceived will almost certainly be influencing the contribution of other sensory systems to balance control. But there is no reason to believe that the vestibular system is special in that it is the only sensory system that influences balance control below perceptual thresholds. Why would one assume that there are no interactions among the various sensory systems contributing to balance in conditions where body motion is not perceived?

The authors' response provides some additional information related to the body sway levels evoked in their 0.2 mA EVS conditions (range 0.27 to 0.38 deg) as compared to what they state were larger sways reported in Peterka (2002). But this is not true. In Peterka (2002) (Figure 4; eyes closed condition) stimulus-evoked RMS sways were about 0.15 deg and 0.25 deg at surface tilt peak-to-peak amplitudes of 0.5 deg and 1 deg. Comparing responses to these two stimulus amplitudes there

was evidence for some proprioceptive/vestibular changes (i.e., frequency response function gains lower and estimated proprioceptive weight lower (implying vestibular weight higher) at the 1 deg compared to 0.5 deg stimulus amplitude) - so there is evidence for vestibular/non-vestibular interactions using stimuli that evoked smaller sway amplitudes than the 0.2 mA stimuli used in the current study.

This reviewer is not understanding the arguments presented related to the Héroux results (which were quite interesting and informative related to recalibration of sensory signals). Those results mainly focus on the recalibration potential of sensory integration. The general notion of Bayesian integration implies that the nervous system will never completely ignore information from a sensory system so the lack of change in sway behavior in the non-coherent conditions could just imply a stable level of the vestibular contribution to balance with the increased sway levels caused by the EVS which could not be completely ignored by the balance control system.

The authors provide an alternative hypothesis that focuses on maintaining signal-to-noise is supported by the similar variability across EVS conditions. But their variability measure is the variability in a narrow region around the stimulus frequency. One could imagine that the nervous system would be concerned with regulating overall variability (as in van der Kooij and Peterka 2011).

Overall, this reviewer does not object to the authors having their own ideas and hypothesis to explain their results, but they need to be more appreciative of the additional discussion presented above and to not discount sensory interactions that can occur below perceptual levels. Indeed, this fits completely with their main point that motor responses occur at stimulus levels that do not involve perception of movement. Additionally, the mention of 'another factor' (sentence beginning on line 651) seems entirely consistent with sensory reweighting ideas.

Other comments:

1. Line 298. I think this should be -0.26 mA, not 0.26 mA.
2. Figure 2. It's very hard to see the difference between the gray and black lines in the middle row of Figure 2. Suggest using color - may red lines instead of gray lines. This color scheme could be extended to the results shown in the 3rd row.
3. Line 410. Confusing to say "for all EVS" when later in the sentence (line 413) it wasn't true for 0.1 Hz.
4. Sentence beginning on line 494. Suggest "Consistent with previously . . . our balance control thresholds to . . ."
5. Figure 5 legend. Suggest beginning on line 523 ". . . stimulus frequency at each stimulus amplitude." Because the figure panels in the top row are clearly labeled, I don't think it's necessary to also add the left panel, middle panel, and right panel descriptions.
6. Line 543. Should be "EVS amplitudes"
7. Lines 545 to 548. Usually when describing a range the order is given as the lowest value to highest value. But I'm not certain it's necessary to mention these ranges since the main points of how they varied with EVS amplitude and frequency were described in the previous sentence and the figure is quite clear.

8. Line 640. Should be ". . . noisy EVS were an order . . .". But this is the sentence where I don't understand the arguments related to the Héroux paper.

9. Sentence beginning line 717. I don't understand this sentence.

Referee #2:

I thank the authors for careful consideration of my feedback on their initial submission and for providing a thorough point-by-point response. I am satisfied that they have made appropriate changes/responses, including additional analysis. I have just the following (very) minor points:

Line 238: reference to Fig. 1A lower panel is no longer correct, as it is now Fig, 1B upper panel

Figure 1: panel B is less clear than the original submission, as now stretched horizontally

Lines 622-623: "(thresholds varied by a factor of 3-9 between studies)" - while I agree with the point being made (i.e. variability between studies and that the current method is objective, not subjective), I still can't figure out exactly where the values (i.e. 3-9) come from. Perhaps stating the range seen in previous studies would improve clarity?

END OF COMMENTS

Québec City, February 24th, 2025

Dear Editor Carson,

We thank you for the opportunity to revise our paper. Please find a revised version of our manuscript entitled "Balance control threshold to vestibular stimuli" that we hope will be suitable for publication in the Journal of Physiology. The modifications within the revised manuscript related to reviewers' comments are denoted by "red text".

EDITOR COMMENTS

Reviewing Editor:

Thank you for considering and responding to the points raised by the reviewers and the changes made have improved the overall quality of the manuscript. You will see that reviewer 1 has raised some points related to the revision and their previous comments. Could you respond to the points being raised and modify the manuscript appropriately or provide the rationale for keeping things as they are.

Also, currently you do not have an acknowledgments section in the manuscript - if appropriate could you include one.

Response: Thank you for the opportunity to respond to the comments raised by both reviewers. We have addressed the main points raised by reviewer 1. Briefly, we modified the text in the revised paper to clarify that the previous text did not imply other senses can only detect whole-body motion when perceived (Introduction) and refined the arguments related to sensory reweighting (Discussion) while acknowledging the current experiment did not alter other senses to test this hypothesis. We also added an acknowledgments section to the manuscript.

REFEREE COMMENTS

Referee #1:

Comments:

This reviewer was generally satisfied with the revisions but there remain a few items that need to be clarified. I continue to agree that this study represents impactful research for the field, the study design and data analysis was very good, and the results were clearly presented.

However, the response to Reviewer #1's first major comment was not entirely satisfying. Specifically, the issue is about whether non-vestibular sensory systems necessarily have to be perceived before they are able to modify the response to EVS. The authors' explanatory response to the comment states an agreement with the reviewer that ". . . there is no reason to think that other sensory systems . . . are not contributing to balance control in the region of very small balance disturbances evoked by low amplitude EVS." But the modified text (beginning on line 132; "Given that . . .") still implies that it would be necessary for body motion be perceived before non-vestibular sources influence balance responses. This reviewer agrees that larger EVFs that induce body motion large enough to be perceived will almost certainly be influencing the contribution of other sensory systems to balance control. But there is no reason to believe that the vestibular system is special in that it is the only sensory system that influences balance control below perceptual thresholds. Why would one assume that there are no interactions

among the various sensory systems contributing to balance in conditions where body motion is not perceived?

The authors' response provides some additional information related to the body sway levels evoked in their 0.2 mA EVS conditions (range 0.27 to 0.38 deg) as compared to what they state were larger sways reported in Peterka (2002). But this is not true. In Peterka (2002) (Figure 4; eyes closed condition) stimulus-evoked RMS sways were about 0.15 deg and 0.25 deg at surface tilt peak-to-peak amplitudes of 0.5 deg and 1 deg. Comparing responses to these two stimulus amplitudes there was evidence for some proprioceptive/vestibular changes (i.e., frequency response function gains lower and estimated proprioceptive weight lower (implying vestibular weight higher) at the 1 deg compared to 0.5 deg stimulus amplitude) - so there is evidence for vestibular/non-vestibular interactions using stimuli that evoked smaller sway amplitudes than the 0.2 mA stimuli used in the current study.

This reviewer is not understanding the arguments presented related to the Héroux results (which were quite interesting and informative related to recalibration of sensory signals). Those results mainly focus on the recalibration potential of sensory integration. The general notion of Bayesian integration implies that the nervous system will never completely ignore information from a sensory system so the lack of change in sway behavior in the non-coherent conditions could just imply a stable level of the vestibular contribution to balance with the increased sway levels caused by the EVS which could not be completely ignored by the balance control system.

The authors provide an alternative hypothesis that focuses on maintaining signal-to-noise is supported by the similar variability across EVS conditions. But their variability measure is the variability in a narrow region around the stimulus frequency. One could imagine that the nervous system would be concerned with regulating overall variability (as in van der Kooij and Peterka 2011).

Overall, this reviewer does not object to the authors having their own ideas and hypothesis to explain their results, but they need to be more appreciative of the additional discussion presented above and to not discount sensory interactions that can occur below perceptual levels. Indeed, this fits completely with their main point that motor responses occur at stimulus levels that do not involve perception of movement. Additionally, the mention of 'another factor' (sentence beginning on line 651) seems entirely consistent with sensory reweighting ideas.

Response: We thank the reviewer for the kind words about the impact of our work, the openness to our ideas and for the comments to help clarify the message.

First, we fully agree with the reviewer that other senses can contribute to balance (and detect whole-body motion) below perceptual thresholds. The sentence highlighted by the reviewer was not meant to convey this idea, but given the confusion, we have modified this sentence to:

“Given that the larger whole-body movements induced by higher EVS currents (Day et al., 1997; Latt et al., 2003; Wardman et al., 2003a) were more likely to be detected by multiple sensors and to potentially evoke (mis)perception of motion (Teasdale et al., 1999; Tisserand et al., 2022), we hypothesized that higher EVS amplitudes would lead to higher balance thresholds to vestibular stimuli.”

We hope this removes any potential confusion.

Next, we have incorporated the text from the reviewer related to the Héroux et al. (2015) paper:

“Also, the stable vestibular contribution to balance induced by noisy EVS an order of magnitude larger than the currents used here (see non-coherent vestibular stimulus in Héroux et al., 2015) and the inference of linear accelerations to EVS showing functioning canal-otolith integration mechanisms (Khosravi- Hashemi et al. 2019) suggest another mechanism should be considered.”

We also moved the text supporting the reweighting idea to earlier in the Discussion and we clearly stated that our experiment was not designed to test directly the sensory reweighting hypothesis because we did not alter other senses. Finally, we recognized that we only quantified variability at the stimulus frequency. Related to this last point, however, we emphasize that there is minimal evidence of nonlinear transmission of EVS to motor responses for stimuli between 0 and 5 mA (i.e., EVS at 0.2 Hz evokes responses mostly at that frequency; Forbes et al. 2014, Hannan et al. 2021), thus supporting our approach.

Regarding the point related to our response to the reviewer (not included in the revised paper), the CoM variability was not a direct measure of whole-body sway evoked by EVS. Indeed, we quantified the overall CoM variability, and the sham condition exhibited an average variability of 0.25 deg. In our experiment, the RMS of the CoM angular body sway for the 0.2 mA EVS amplitude across participants was 0.34°, 0.38°, 0.27° and 0.36° for the 0.1, 0.2, 0.5, and 1 Hz frequencies. Thus, the increase in RMS of the CoM angular body sway from sham to 0.2 mA EVS was between 0.02 and 0.13 deg.

Other comments:

1. Line 298. I think this should be -0.26 mA, not 0.26 mA.

Response: This has been corrected.

2. Figure 2. It's very hard to see the difference between the gray and black lines in the middle row of Figure 2. Suggest using color - may red lines instead of gray lines. This color scheme could be extended to the results shown in the 3rd row.

Response: As suggested by the reviewer, we added color to contrast sham EVS from the EVS time series.

3. Line 410. Confusing to say "for all EVS" when later in the sentence (line 413) it wasn't true for 0.1 Hz.

Response: We thank the reviewer for this suggestion and changed this sentence to:

“Compared to sham EVS, these larger peaks in spectral amplitude were significant for EVS frequencies larger than 0.1 Hz (EVS 0.2 mA: $p = 0.04$, $p = 0.009$ and, $p = 0.02$; EVS 0.4 mA: $p = 0.009$, $p = 0.004$, $p = 0.006$; EVS 0.6 mA: $p = 0.005$, $p = 0.006$, $p = 0.02$) but not for the 0.1 Hz EVS frequency ($p = 0.25$, $p = 0.20$, and $p = 0.17$ for EVS of 0.2, 0.4 and 0.6 mA). ”

4. Sentence beginning on line 494. Suggest "Consistent with previously . . . our balance control thresholds to . . ."

Response: We thank the reviewer and accepted this suggestion.

5. Figure 5 legend. Suggest beginning on line 523 ". . . stimulus frequency at each stimulus amplitude." Because the figure panels in the top row are clearly labeled, I don't think it's necessary to also add the left panel, middle panel, and right panel descriptions.

Response: We removed this description, as suggested by the reviewer.

6. Line 543. Should be "EVS amplitudes"

Response: This has been corrected.

7. Lines 545 to 548. Usually when describing a range the order is given as the lowest value to highest value. But I'm not certain it's necessary to mention these ranges since the main points of how they varied with EVS amplitude and frequency were described in the previous sentence and the figure is quite clear.

Response: To avoid confusion, we added in the text that the reported data are in order of EVS frequencies (i.e. *from 0.1 to 1 Hz*). We prefer to report these data for further comparison with future studies and reporting the range in the reverse order would be conflict with the figures that show the equivalent angular velocity and linear acceleration thresholds decrease as a function of EVS frequencies.

8. Line 640. Should be ". . . noisy EVS were an order . . .". But this is the sentence where I don't understand the arguments related to the Héroux paper.

Response: As mentioned in the main point above, we have revised this sentence according to the suggestion from the reviewer. The revised sentence now reads:

"Also, the stable vestibular contribution to balance induced by noisy EVS an order of magnitude larger than the currents used here (see non-coherent vestibular stimulus in Héroux et al., 2015) and the inference of linear accelerations to EVS showing functioning canal-otolith integration mechanisms (Khosravi- Hashemi et al. 2019) suggest another mechanism should be considered."

9. Sentence beginning line 717. I don't understand this sentence.

Response: We modified the sentence to:

"Given that the balance thresholds from 6 and 3 participants were larger than the applied 0.2 mA EVS at the 0.1 and 0.2 Hz frequencies, we recommend using 0.2 mA stimuli at 0.5 Hz to assess balance thresholds to vestibular stimuli in healthy controls."

Hence, the thresholds calculated were larger than the 0.2 mA EVS amplitude for 6 and 3 participants when exposed to EVS frequencies of 0.1 and 0.2 Hz. This support our recommendation of using a 0.2 mA EVS at 0.5 Hz because the thresholds from all participants were smaller than the applied EVS amplitude.

Referee #2:

I thank the authors for careful consideration of my feedback on their initial submission and for providing a thorough point-by-point response. I am satisfied that they have made appropriate changes/responses, including additional analysis. I have just the following (very) minor points:

We thank the reviewer for their constructive feedback.

Line 238: reference to Fig. 1A lower panel is no longer correct, as it is now Fig, 1B upper panel

Response: Thank you, this has been corrected.

Figure 1: panel B is less clear than the original submission, as now stretched horizontally

Response: We have modified the figure to better illustrate the threshold on the x-axis (i.e., added an arrow with the word threshold). We removed the horizontal stretching, as suggested.

Lines 622-623: "(thresholds varied by a factor of 3-9 between studies)" - while I agree with the point being made (i.e., variability between studies and that the current method is objective, not subjective), I still can't figure out exactly where the values (i.e., 3-9) come from. Perhaps stating the range seen in previous studies would improve clarity?

Response: We thank the reviewer for this comment. We rephrased this sentence to avoid confusion.

We change the sentence to:

"Although we lack a definitive explanation for the variability in standing participants' thresholds between our estimates (0.09-0.57 mA) and those reported previously (ranging from 0.2-1.94 mA; Bent et al., 2000 and Mikhail et al., 2021), we propose that our methods reflect thresholds of the vestibular control of balance by relying on signal statistics and signal detection theory that avoid subjective assessments of whole-body motion."

References:

Forbes, P. A., Dakin, C. J., Geers, A. M., Vlaar, M. P., Happee, R., Siegmund, G. P., ... & Blouin, J. S. (2014). Electrical vestibular stimuli to enhance vestibulo-motor output and improve subject comfort. PLoS One, 9(1), e84385.

Hannan, K. B., Todd, M. K., Pearson, N. J., Forbes, P. A., & Dakin, C. J. (2021). Absence of nonlinear coupling between electric vestibular stimulation and evoked forces during standing balance. Frontiers in human neuroscience, 15, 631782.

Dear Professor Simoneau,

Re: JP-RP-2025-288016R2 "Balance control threshold to vestibular stimuli" by Martin Simoneau, Mujda Nooristani, and Jean-Sébastien Blouin

We are pleased to tell you that your paper has been accepted for publication in The Journal of Physiology.

Yours sincerely,

Richard Carson
Senior Editor
The Journal of Physiology

If you would like to receive our 'Research Roundup', a monthly newsletter highlighting the cutting-edge research published in The Physiological Society's family of journals (The Journal of Physiology, Experimental Physiology, Physiological Reports, The Journal of Nutritional Physiology and The Journal of Precision Medicine: Health and Disease), please click this link, fill in your name and email address and select 'Research Roundup':

<https://www.physoc.org/journals-and-media/membernews>

- **TRANSPARENT PEER REVIEW POLICY:** To improve the transparency of its peer review process, The Journal of Physiology publishes online as supporting information the peer review history of all articles accepted for publication. Readers will have access to decision letters, including Editors' comments and referee reports, for each version of the manuscript as well as any author responses to peer review comments. Referees can decide whether or not they wish to be named on the peer review history document.
- You can help your research get the attention it deserves! Check out Wiley's free Promotion Guide for best-practice recommendations for promoting your work at: www.wileyauthors.com/eeo/guide. You can learn more about Wiley Editing Services which offers professional video, design, and writing services to create shareable video abstracts, infographics, conference posters, lay summaries, and research news stories for your research at: www.wileyauthors.com/eeo/promotion.
- **IMPORTANT NOTICE ABOUT OPEN ACCESS:** To assist authors whose funding agencies mandate public access to published research findings sooner than 12 months after publication, The Journal of Physiology allows authors to pay an Open Access (OA) fee to have their papers made freely available immediately on publication.

EDITOR COMMENTS

Reviewing Editor:

Thank you for considering the reviewer's comments and addressing them in your revised manuscript.

REFEREE COMMENTS

Referee #1:

I have reviewed the changes made in this second revision of the manuscript and am satisfied that the changes have addressed my previous concerns. The study represents impactful research in the field. The study is well designed and results are clearly presented.